# Endothelial junctional membrane protrusions serve as hotspots for neutrophil transmigration

Janine JG Arts[1,2], Eike K Mahlandt[2], Max LB Grönloh[1,2], Lilian Schimmel[1,2,3],
Ivar Noordstra[3], Emma Gordon[3], Abraham CI van Steen[1], Simon Tol[1],
Barbara Walzog[4], Jos van Rijssel[1], Martijn A Nolte[1], Marten Postma[2],
Satya Khuon[5], John M Heddleston[5,6], Eric Wait[5], Teng Leong Chew[5],
Mark Winter[7], Eloi Montanez[8], Joachim Goedhart[2], Jaap D van Buul[1,2]*

[1]Molecular Cell Biology Lab at Dept. Molecular Hematology, Sanquin Research and
Landsteiner Laboratory, Amsterdam, Netherlands; [2]Leeuwenhoek Centre for
Advanced Microscopy (LCAM), section Molecular Cytology at Swammerdam
Institute for Life Sciences (SILS) at University of Amsterdam, Amsterdam,
Netherlands; [3]Division of Cell and Developmental Biology, Institute for Molecular
Bioscience, The University of Queensland, BrisbaneQLD, Australia; [4]Department of
Cardiovascular Physiology and Pathophysiology, Walter Brendel Center of
Experimental Medicine, Biomedical Center, Ludwig-Maximilians-Universität
München, Planegg-Martinsried, Germany; [5]Advanced Imaging Center at Janelia
Research Campus, Howard Hughes Medical Institute, Ashburn, United States;
[6]Microscopy Facility at the Cleveland Clinic Florida Research and Innovation Center,
Port St. Lucie, United States; [7]Zuckerman Postdoctoral Fellow, Department of
Marine Sciences, University of Haifa, Haifa, Israel; [8]Department of Physiological
Sciences, Faculty of Medicine and Health Sciences, University of Barcelona,
Barcelona, Spain

*For correspondence:
j.vanbuul@sanquin.nl

**Competing interests:** The
authors declare that no
competing interests exist.

**Reviewing editor:** Carla V
Rothlin, Yale School of Medicine,
United States

**Abstract** Upon inflammation, leukocytes rapidly transmigrate across the endothelium to enter
the inflamed tissue. Evidence accumulates that leukocytes use preferred exit sites, alhough it is not
yet clear how these hotspots in the endothelium are defined and how they are recognized by the
leukocyte. Using lattice light sheet microscopy, we discovered that leukocytes prefer endothelial
membrane protrusions at cell junctions for transmigration. Phenotypically, these junctional
membrane protrusions are present in an asymmetric manner, meaning that one endothelial cell
shows the protrusion and the adjacent one does not. Consequently, leukocytes cross the junction
by migrating underneath the protruding endothelial cell. These protrusions depend on Rac1 activity
and by using a photo-activatable Rac1 probe, we could artificially generate local exit-sites for
leukocytes. Overall, we have discovered a new mechanism that uses local induced junctional
membrane protrusions to facilitate/steer the leukocyte escape/exit from inflamed vessel walls.

## Introduction

The current paradigm of leukocyte transendothelial migration (TEM) comprises leukocyte rolling,
arrest, crawling, firm adhesion, and diapedesis (*Alon and van Buul, 2017*; *Butcher, 1991*;
*Muller, 2016*; *Nourshargh and Alon, 2014*; *Springer, 1994*; *Vestweber, 2015*). The latter step
occurs either through the endothelial junctions, known as the paracellular route or through the endo-
thelial cell body, called the transcellular route (*Carman, 2009*; *Wittchen, 2009*). Interestingly, some

locations seem to favor the migration of multiple leukocytes that breech the endothelium in rapid succession (*Abtin et al., 2014*). Prior to exiting the circulation, leukocytes change their crawling morphology to a round appearance, indicating that at exit-sites differential and local regulation is required (*Shulman et al., 2009*). However, what determines where leukocytes take the exit remains elusive.

When leukocytes crawl on the endothelial monolayer, it seems they are looking for the perfect spot to leave the circulation. As if there would be a preferred 'hotspot' for these leukocytes to cross the endothelium. So far, several key principles have been suggested that may define local endothelial exit-sites for leukocytes (*Muller, 2015*; *Schimmel et al., 2017*; *Vestweber, 2015*). First, leukocytes are attracted toward an optimal concentration of chemokines (*chemotaxis*), density of adhesion molecules (*haptotaxis*) or cellular stiffness (*durotaxis*). Recently, it has been shown that perivascular macrophages can secrete chemokines that may cause local queues for neutrophil diapedesis in vivo (*Abtin et al., 2014*). Similar ideas have been suggested for *haplotaxis*: integrin ligands presented at the apical surface of the endothelium that regulate leukocyte behaviour (*Shulman et al., 2009*). For example, high surface levels of ICAM-1 induce a transition from paracellular to robust transcellular migration, while intermediate levels favor the paracellular route (*Abadier et al., 2015*; *Yang et al., 2005*), although the underlying mechanism is unclear. Data from our lab (*Heemskerk et al., 2016*; *Timmerman et al., 2016*; *van Buul et al., 2007*; *van Buul et al., 2010*) and that of others (*Adamson et al., 1999*; *Barreiro et al., 2002*; *Barreiro et al., 2008*; *Carman and Springer, 2004*; *Lyck and Enzmann, 2015*; *Muller, 2015*; *Vestweber, 2015*) indicate a dynamic role for the endothelial actin cytoskeleton in controlling the factors described above. Recently, Hyun and colleagues showed the existence of two different hotspots during extravasation in vivo (*Hyun et al., 2019*), one of which is located on the endothelium.

Perturbation of the endothelial actin cytoskeleton drastically influences the phenotype of the adherent leukocyte: blocking actin polymerization of the endothelium prevents the leukocyte to properly adhere and spread and consequently, leukocyte TEM is critically impaired (*Barreiro et al., 2002*; *Carman et al., 2003*; *Carman and Springer, 2004*; *Schimmel et al., 2018*; *van Buul et al., 2010*). On the other hand, once a leukocyte has adhered to the endothelium, F-actin is locally (i.e. underneath the adherent leukocyte) depolymerized (*Isac et al., 2011*) These findings suggest that the endothelium can orchestrate local leukocyte diapedesis by remodeling its actin cytoskeleton at the subcellular level. Thus, understanding how actin-regulatory protein complexes orchestrate TEM under inflammation may thus be key to discover the mechanisms of the endothelium that drive local leukocyte exit.

Our observations underscore that leukocyte diapedesis is not a random event but occurs at pre-defined exit-sites. As TEM is a very rapid process (leukocytes cross the endothelium within 120 s), we used lattice light sheet microcopy, allowing high-speed imaging in three dimensions in time with high resolution but low toxicity to monitor TEM in detail. We discovered a novel mechanism how the endothelium generates local exit-sites for leukocytes to leave the vasculature. By asymmetrically inducing Rac1- and Arp3-dependent apical membrane protrusions at cell-cell junction regions that express ACKR1, ICAM-1, and PECAM-1, the endothelium guides leukocytes to the exit-side. Additionally, these protrusions are also found in vivo, and local activation of Rac1, using a photo-activatable Rac1 probe drives leukocyte exit on demand. Our work identified so-called endothelial junctional membrane protrusions (JMPs), a novel molecular mechanism that allows local steering of leukocyte TEM at the vascular level and offers new therapeutic targets to locally inhibit or enhance leukocyte extravasation.

## Results

### Endothelial cells display junctional membrane protrusions

When neutrophils crawl on an endothelial surface under inflammatory conditions, they are searching for an exit-point. Although most of the neutrophils seems to transmigrate at the first junction, their migratory behavior is more complex, and consists of walks along a junction and turns after crossing a junction (*Schenkel et al., 2004*). Using transendothelial migration (TEM) under flow assays (*Heemskerk et al., 2016*), we observed that although most neutrophils transmigrated at the first cell-cell junction they encounter (*Figure 1A*), only half of them transmigrated immediately upon

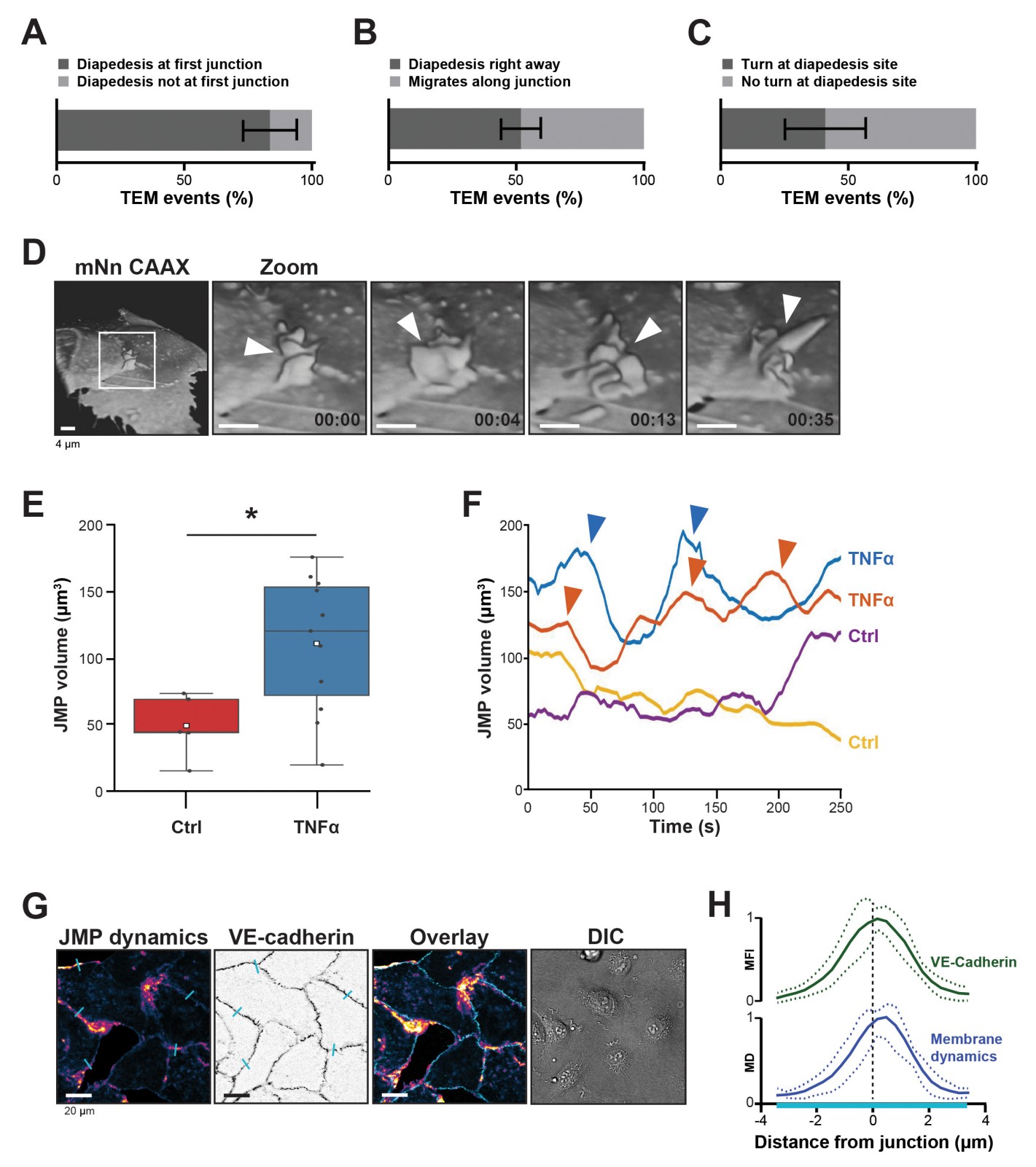

**Figure 1.** Presence of endothelial junctional membrane protrusions. (**A**) Quantification in percentage of diapedesis events at first junction neutrophils encounter (dark bar) versus neutrophils that cross more than one junction before transmigrating (light bar). N=3, on average 13 TEM events per experiment. Mean ± standard deviation (SD). (**B**) Quantification in percentage of diapedesis events directly upon encountering a junction region (dark bar) versus neutrophils that migrate along such junction region prior to diapedesis (light bar). N=3, on average 13 TEM events per experiment. Mean ±

*Figure 1 continued on next page*

*Figure 1 continued*

SD. (C) Quantification in percentage of diapedesis events at junction regions for neutrophils that cross straight through (dark bar) versus neutrophils that make a turn at junction regions prior to diapedesis (light bar). N=3, on average 13 TEM events per experiment. Mean ± SD. (D) 3D-view stills from mNeonGreen-CAAX transfected HUVECs showing presence of junctional membrane protrusions, indicated by white arrowheads. Time indicated in seconds in lower right corner. Bar, 4 µm. (E) Boxplots of the volume average of 3D junctional membrane protrusions in time upon TNF-α treatment (blue) versus control (red). White square represents mean. Mann-Whitney U-test: *p=0.0275. (F) Quantification of the dynamics of JMP volume of control (yellow and purple lines) versus 20 hr TNF-α-treated (blue and orange lines) endothelial cells as indicated. two representative lines per condition are shown. JMPs from TNF-treated ECs show more fluctuations in volume than JMPs from control cells as indicated by arrowheads. (G) Junctional membrane dynamics map in pseudo-colors, warm colors indicate high membrane dynamics, cold colors indicate low membrane dynamics (i), VE-cadherin staining (ii), overlay (iii), and DIC (iiii). Turquoise lines indicate sites of line scans for quantification. Bar, 20 µm. (H) Quantification of normalized fluorescence intensity (MFI) of blue lines as indicated in E by line scan analysis of VE-cadherin (green) and normalized value from membrane dynamics map (MD) (dark blue). Green and dark blue lines represent mean of five independent junctions, dotted lines show SD.

The online version of this article includes the following video and figure supplement(s) for figure 1:

**Figure supplement 1.** JMP dynamics.
**Figure 1—video 1.** Transmigration of neutrophils across TNFα-treated endothelial monolayers.
https://elifesciences.org/articles/66074#fig1video1
**Figure 1—video 2.** JMP dynamics.
https://elifesciences.org/articles/66074#fig1video2
**Figure 1—video 3.** JMP volume and area quantification of 20 hr TNFα-treated endothelial cells.
https://elifesciences.org/articles/66074#fig1video3
**Figure 1—video 4.** JMP volume and area quantification of non-treated endothelial cells.
https://elifesciences.org/articles/66074#fig1video4

arrival to that junction (*Figure 1B*). The other half migrated laterally along this junction for at least 20 s before crossing. In fact, approximately 40% of these neutrophils changed direction upon diapedesis, meaning that once arrived at the cell-cell junction, these neutrophils migrated on top of that junction, turned around and exited by migrating underneath the endothelial cell that they previously crawled on (*Figure 1C*), in line with *Schenkel et al., 2004*. The findings presented here confirm that under physiological flow conditions, most neutrophils search for specific sites on the endothelium to exit (*Figure 1—video 1*).

To study the morphology and dynamics of the endothelial cell membranes in full detail, we used lattice light sheet microscopy (LLSM). This technique allows for extremely fast imaging in 3D in time with high resolution and low phototoxicity (*Chen et al., 2014*). Membranes of human umbilical vein endothelial cells (HUVECs) were labeled with a fluorescently tagged membrane marker (mNeon-Green-CAAX). 3D reconstructions revealed that the endothelium extended membrane protrusions apically into the lumen (*Figure 1D* and *Figure 1—figure supplement 1A*). These protrusions showed a highly dynamic appearance and various morphological shapes, ranging from finger-like to flap-like structures and are not restricted to one junction (*Figure 1—video 2*). As these protrusions appeared at junction regions, we termed them junctional membrane protrusions (JMPs). To study if JMPs are evoked by inflammation, we developed a robust and unbiased algorithm that allowed us to follow JMPs in 3D in time and quantify JMP volume, dynamics, and area over time (*Figure 1—figure supplement 1B* and *Figure 1—videos 3* and *4*). These data revealed that short-term, that is, 4 hr (not shown), and long-term TNFα treatment, that is, 20 hr, to the same extend increased the overall volume of the JMPs as well as the dynamics of these JMPs, as well as the area of the JMP on the endothelium, although the latter one was not significant (*Figure 1E–F* and *Figure 1—figure supplement 1C–D*).

The LLSM data provided highly detailed images of protrusion structures but is not suitable for large number of samples and testing of multiple experimental conditions. Therefore, to simplify the observation of JMP dynamics, we developed a method that enables the quantification of JMP dynamics with widefield microscopy imaging data of HUVECs with a fluorescent membrane marker. The rational of the image analysis was that highly dynamic membranes will display larger fluctuations in fluorescence intensity as compared to static membranes. *Figure 1G* shows an example of such a JMP dynamics in time map and showed that JMPs colocalized with VE-cadherin (*Figure 1H*).

We additionally checked if other inflammatory stimuli, that is, LPS, INFγ, and IL1β, changed the dynamic rate of JMPs compared to TNFα. Using the simplified algorithm to generate a membrane

dynamics map, we quantified JMPs by simply taking the mean value of this image (for details see Materials and methods section). We found that all inflammatory stimuli tested showed the same JMP dynamics (*Figure 2—figure supplement 1A*). Also, when we tested JMP dynamics in endothelial cells that were isolated from different origin, we did not find differences (*Figure 2—figure supplement 1B*). As transmigration events occurred under flow conditions, we tested if flow altered JMP dynamics. Indeed, short-term flow for 5 or 20 min did increase JMP activity, whereas exposure of the endothelial cells to longer periods of flow, that is, 30–60 min, reduced JMP dynamics back to basal levels (*Figure 2—figure supplement 1C*). We did not find any preference of JMP dynamics at the upstream or downstream side of the flow direction (*Figure 2—figure supplement 1C*). Thus, these data show that inflammatory stimuli promote the dynamics but not the actual appearance or size of JMPs.

## Neutrophils prefer local exit-sites

Given the spatial heterogeneity of the membrane protrusions, we wondered how JMPs are related to the neutrophil exit sites. To investigate this, mNeonGreen-CAAX transduced HUVECs were used to study neutrophil TEM under flow conditions. We found that most neutrophils preferred exit-sites with high-membrane activity (*Figure 2A and B*). As about 50% of the neutrophils showed prolonged crawling time prior to diapedesis (*Figure 1B*), we compared JMP dynamics of the first junctional region that a crawling neutrophil encountered to the JMP dynamics at sites where the neutrophil underwent diapedesis. In line with the idea that neutrophils preferred JMPs, we found that JMP dynamics were increased at sites where neutrophils transmigrated compared to sites where neutrophils migrated along the junction but did not exit (*Figure 2C*). Moreover, we found that neutrophils preferred to initiate diapedesis by crawling towards a JMP of the neighboring endothelial cell. Consequently, the neutrophil used this JMP to exit and continued crawling underneath the endothelial cell that generated the JMP (*Figure 2D* and *Figure 2—figure supplement 1E–F*). Interestingly, we observed that once a neutrophil used a JMP to cross the endothelium, other neutrophils followed this path and used the same exit-site (*Figure 2E*). This was in line with an increase in endothelial JMP dynamics at exit-sites after the first neutrophil crossed (*Figure 2F*), in line with a recent publication by *Hyun et al., 2019*. These data suggest that endothelial JMPs can function as exit-recognition sites that are reinforced by transmigration.

## PECAM-1-positive JMPs express ICAM-1, ACKR1 and found in vivo

To study the JMP phenotype in more detail, we used LLSM imaging and found that VE-cadherin localized at the basolateral area of JMPs but is not present on the actual JMP (*Figure 3A*). In contrast, junctional PECAM-1/CD31 was present on JMPs (*Figure 3B* and *Figure 3—figure supplement 1A*). Using a 3D quantification algorithm, we found that PECAM-1 covered at least 50% of the surface of the JMP (*Figure 3C*). To examine whether JMPs are found in vivo and are correlated to TEM events, we used the cremaster muscle as a well-established in vivo model. The cremaster muscle of C57BL/6 mice were treated with TNF/IL1β and subsequently stained for PECAM-1 and neutrophil-specific coronin1 (*Pick et al., 2017*). We found that PECAM-1-rich membrane protrusions can also be found in vivo, although we cannot quantify them as JMPs based on these images. As with the in vitro data, PECAM-1 covered the apical protrusions that surrounded adherent neutrophils in vivo (*Figure 3D*). Thus, these findings show the presence of apical, PECAM-1-decorated endothelial protrusions under inflammation conditions both in vitro and in vivo.

The atypical chemokine receptor 1 (ACKR1), also referred to as Duffy antigen receptor for chemokines (DARC/CD234), was recently reported to be present at endothelial cell-cell junctions and involved in leukocyte TEM (*Girbl et al., 2018*; *Thiriot et al., 2017*). To examine the location of ACKR1, we expressed an mNeonGreen-tagged variant of ACKR1 in HUVECs together with mScarlet-I-CAAX to label the membrane and cultured the cells in a mosaic manner to a full monolayer. We observed that ACKR1 is expressed at CAAX-identified JMPs (*Figure 3E*, *Figure 3—figure supplement 1B–C*), suggesting that chemokines can be presented by JMPs. To identify the role of JMPs in the adhesion stage of TEM, we co-expressed ICAM-1 mScarlet-I and mNeonGreen-CAAX in HUVECs. Like ACKR1, ICAM-1 localized to JMPs (*Figure 3F*, *Figure 3—figure supplement 1D–E*). Interestingly, when measuring the fluorescence at JMPs using a profile plot, we found that fluorescence of the proteins detected at JMPs increased to the same extend as the membrane marker

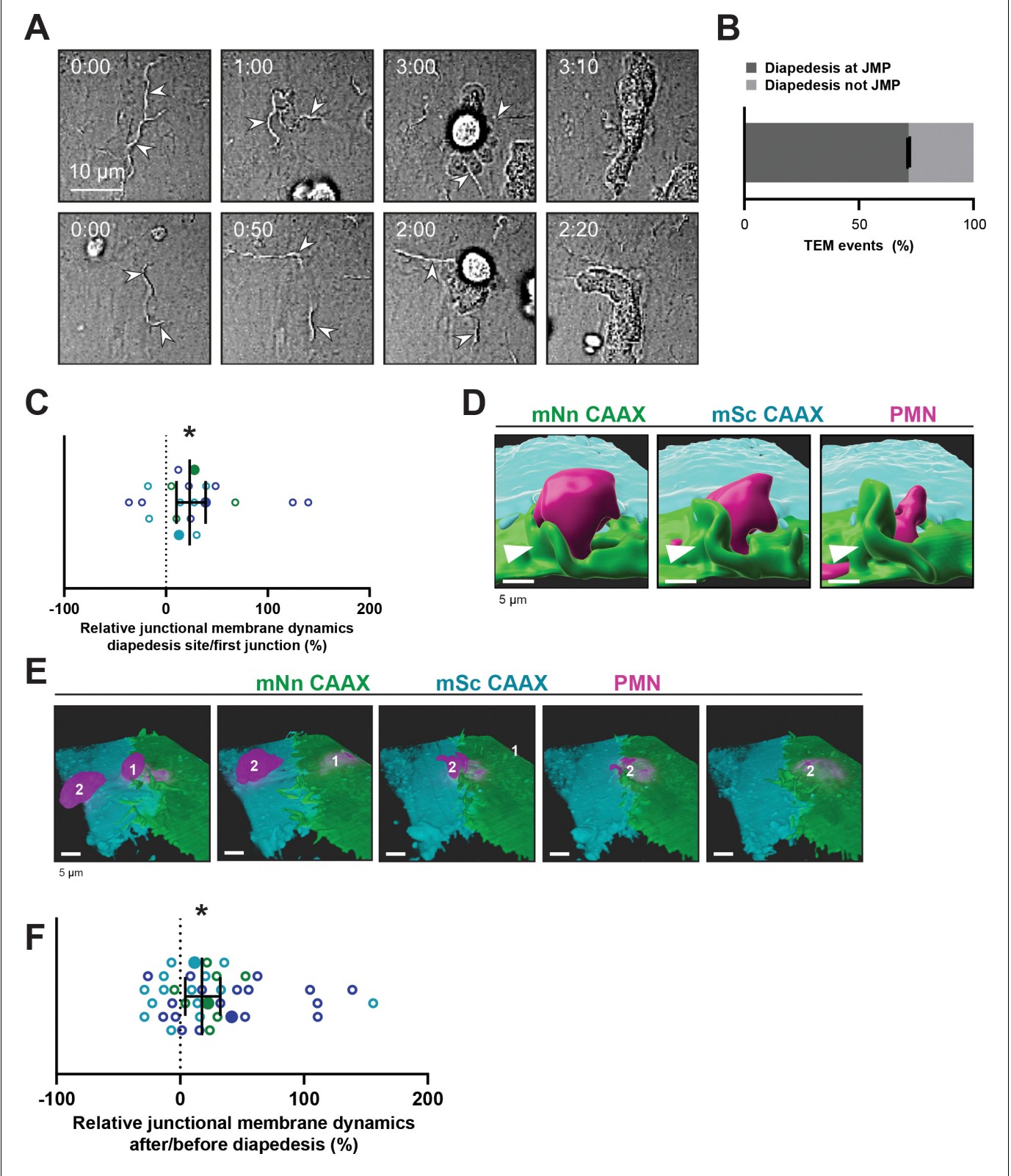

**Figure 2.** Neutrophils use junctional membrane protrusions as diapedesis site. (**A**) Stills from two time-lapse movies showing PMN transendothelial migration (TEM) under flow showing presence of endothelial membrane ruffles, indicated by white arrowheads. The membrane ruffles are present at the site of diapedesis already before PMN adhesion and indicate the spot where the PMN will breach the EC layer. Bar, 10 µm. (**B**) Quantification of the number of TEM events that show elevated junctional membrane dynamics prior to PMN TEM at site of diapedesis. N=3, on average >10 TEM events

*Figure 2 continued on next page*

*Figure 2 continued*

per experiment. (C) Ratio membrane dynamics at diapedesis site and membrane dynamics at site where a neutrophil first encounters. Open dots are individual data points from three independent experiments (three colors). Filled dots are means from three experiments. Median with 95% confidence interval (CI) are shown. One-sample Wilcoxon test: *p=0.0305. (D) 3D image stills using Imaris rendering software from two ECs (green/turquoise) and PMN (magenta) showing junctional membrane protrusion at the diapedesis site. Bar, 5 µm. (E) 3D view image from two endothelial cells (green/turquoise) and PMN (magenta, labeled #1) showing a second PMN (labeled #2) transmigrating at the same diapedesis site as the first neutrophil (#1). Bar, 5 µm. (F) Ratio JMPs at the diapedesis after and before TEM showing an increase in endothelial membrane dynamics after diapedesis. Open dots are individual data points from three independent experiments (three colors). Filled dots are means from three experiments. Median with 95% CI is shown. One-sample Wilcoxon test: *p=0.0006.

The online version of this article includes the following figure supplement(s) for figure 2:

**Figure supplement 1.** Different stimuli promote JMP dynamics.

CAAX (*Figure 3—figure supplements 1B and D*). Based on these data, we concluded that ICAM-1 and ACKR1 are not enriched at JMPs. These results suggest that JMPs may be involved in two key steps of TEM. First by presenting ACKR1 as a chemokine presenter to the crawling leukocytes (*Pruenster et al., 2009*) and second, by presenting endothelial cell adhesion molecules for the leukocyte integrins to start diapedesis.

## Membrane protrusions are regulated by the actin cytoskeleton

Data from our lab and that of others (*Heemskerk et al., 2016*; *van Buul et al., 2010*; *van Buul et al., 2007*; *Adamson et al., 1999*; *Barreiro et al., 2002*; *Barreiro et al., 2008*; *Carman and Springer, 2004*; *Lyck and Enzmann, 2015*; *Muller, 2015*; *Vestweber, 2015*) indicate a prominent role for the endothelial actin cytoskeleton in the regulation of TEM. To study the role of the actin cytoskeleton in JMPs, we used LifeAct-expressing endothelial cells and LLSM imaging, which revealed that JMPs are rich in F-actin (*Figure 4A*). To investigate if these structures are also present in vivo, we used Lifeact-GFP knock-in mice (*Fraccaroli et al., 2012*) and studied the inflamed cremaster muscle vessel morphology. As Lifeact-EGFP is mainly expressed in the vascular endothelium and not in the leukocytes, allowing imaging of the endothelial actin cytoskeleton with excellent contrast (*Fraccaroli et al., 2012*), we used these mice to study the presence of luminal membrane protrusions. In vivo time lapse imaging showed actin-rich vascular structures that protruded apically and were associated with TEM events (*Figure 4B*).

As the Arp2/3 complex is involved in actin nucleation, branching and lamellipodia formation (*Goley and Welch, 2006*), we hypothesized that its activity is required for the dynamics of JMPs. Indeed, treatment of endothelial cells with the Arp2/3 inhibitor CK-666 reduced JMP dynamics (*Figure 4C–D*). Next, we studied the functional consequences of perturbing F-actin branching on neutrophil TEM. However, the use of inhibitors can be problematic in assays that include two different cell types: inhibitors may diffuse out of the endothelial cells and affect migration motility of neutrophils under flow conditions. Therefore, we silenced Arp3 in endothelial cells using shRNA (*Figure 4—figure supplement 1A–B* and *Figure 4—source data 1*). Indeed, silencing Arp3, with two independent shRNAs, resulted in reduced number of neutrophils that crossed the endothelial monolayer under flow conditions (*Figure 4E* and *Figure 4—figure supplement 1C*). To specifically quantify membrane dynamics in Arp3-deficient endothelial cells, we co-expressed the Arp3 shRNA and mNeonGreen-CAAX from one plasmid, assuring that CAAX-expressing cells were indeed silenced for Arp3 (*Figure 4—figure supplement 1D–E*). Endothelial cells that were silenced for Arp3 showed reduced JMP dynamics (*Figure 4F* and *Figure 4—figure supplement 1F*). These data underscore the importance of actin nucleation by the Arp2/3 complex for JMP formation and neutrophil TEM.

## High Rac1 activity correlates with JMP dynamics and diapedesis exit-sites

Small GTPases of the Rho family, in particular Rac1, are well recognized for their regulatory role in the formation of lamellipodia (*Hall, 2005*; *Heemskerk et al., 2014*) and Arp2/3 activity (*Goley and Welch, 2006*). To test if Rac1 is required for JMPs, we used a Rac1 inhibitor (EHT1864) and found that JMP dynamics were reduced (*Figure 5A* and *Figure 5—figure supplement 1A*). Next, we used FRET-based DORA Rac1 biosensor to measure Rac1 activity at JMPs (*Timmerman et al., 2016*;

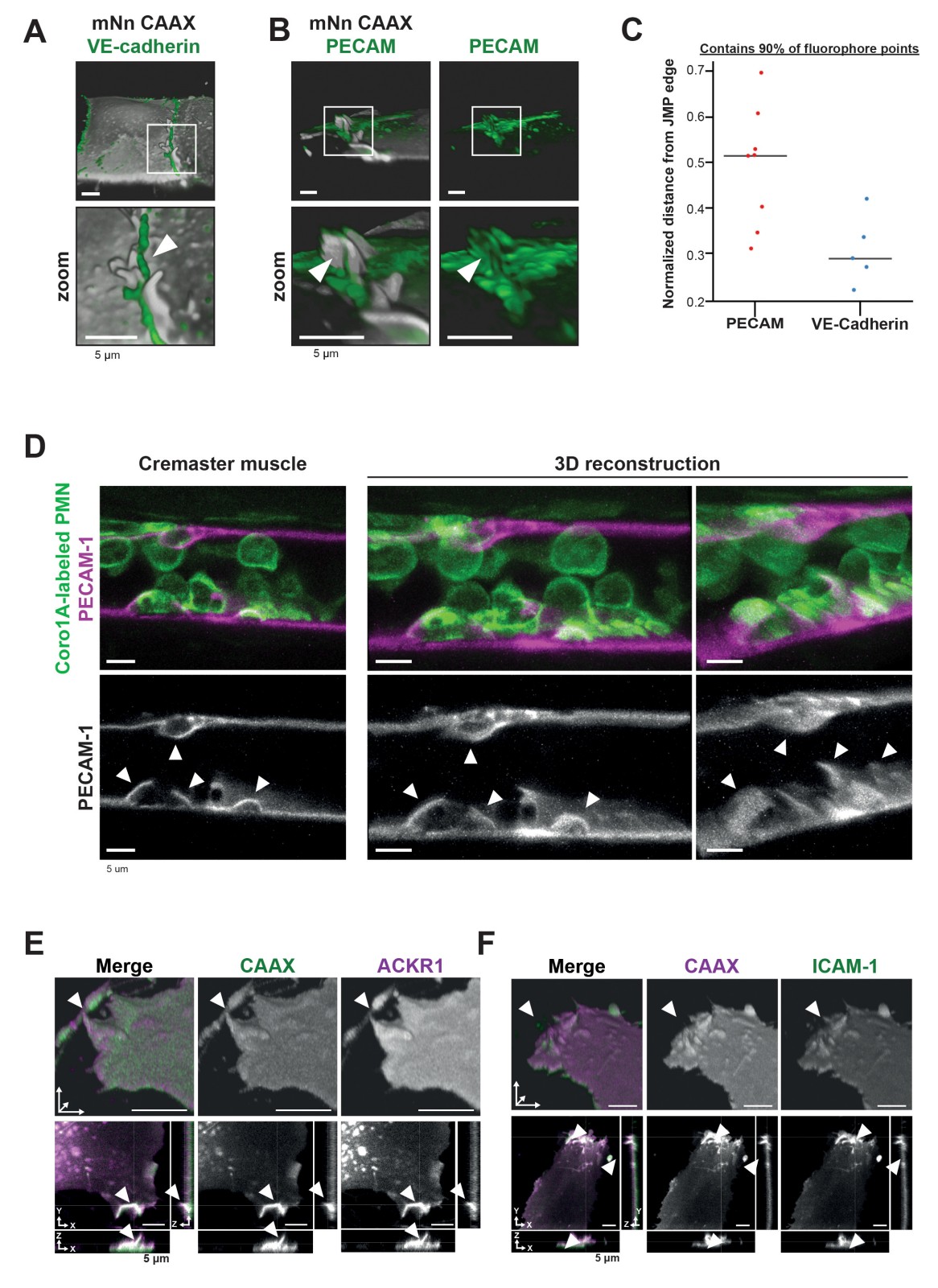

**Figure 3.** PECAM-1-positive junctional membrane protrusions express ICAM-1, ACKR1 and found in vivo. (**A**) LLSM stills from HUVECs expressing membrane-bound CAAX (gray) stained with a directly conjugated VE-cadherin antibody (green). White box indicates zoom region, displayed below. Arrowhead points at JMP that does not overlap with the VE-cadherin staining. Bar, 5 μm. (**B**) LLSM stills from HUVECs expressing the CAAX membrane label (grays) stained with a directly conjugated PECAM-1antibody (green). White boxes indicate zoom regions. Arrowhead points at JMP that show

*Figure 3 continued on next page*

Figure 3 continued

overlap with the PECAM-1 staining. Bar, 5 µm. (C) Quantification of LLSM images of fluorescent coverage of PECAM-1 and VE-cadherin on CAAX-positive JMPs, based on 90% of all available fluorophore points on the volume measurements of the JMPs. Dots indicate one experiment in which one individual JMP is measured. (D) Confocal intravital microscopy of 20–80 µm diameter cremasteric venules in mice with Coro1A stained neutrophils in green immunostained in vivo for EC junctions by intrascrotal injections of fluorescent-labeled PECAM-1 (red) and stimulated for four hours with IL-1β and TNF-α. Fixed images in top row show presence of PECAM-1-positive membrane protrusions that surround adherent neutrophils in green. Two images on the right show reconstruction. Lower row shows PECAM-1 staining in white only. Arrows show presence of PECAM-1-positive membrane protrusions. Scale bar, 5 µm. (E) Stills from two endothelial cells expressing mScarlet-I-CAAX (green) and mNeonGreen ACKR1 (magenta) showing ACKR1-containing JMPs as indicated by white arrowheads. Of note, the endothelium is grown to a full monolayer but untransfected cells are not detected, as they do not express any FP. Bar, 5 µm. (F) Stills from two endothelial cells expressing mNeonGreen-CAAX (green) and ICAM-1 mScarlet-I (magenta) showing ICAM-1-containing JMPs as indicated by white arrowheads. Bar, 5 µm.

The online version of this article includes the following figure supplement(s) for figure 3:

**Figure supplement 1.** JMPs in vitro and in vivo.

Timmerman et al., 2015). To this end, endothelial cells were transfected with the Rac1 biosensor and with mScarlet-I-CAAX. The FRET ratio was used as a read-out of Rac1 activity, and it was measured at JMP regions and compared with the average ratio at the full cell junctional region (for details see Materials and methods section). This analysis showed that JMPs displayed a significant increase in Rac1 activation (*Figure 5B–C* and *Figure 5—figure supplement 1B–C*). To correlate Rac1 activation with neutrophil diapedesis, we mapped the areas where neutrophils crossed the endothelium to the FRET ratio of the Rac1 biosensor. We found that neutrophils preferred to cross the endothelium at sites with high FRET ratio (*Figure 5D–E* and *Figure 5—video 1*). However, no change in Rac1 activity was detected at regions where neutrophils adhered and crawled on the surface of the endothelium (*Figure 5—figure supplement 1D–E*), indicating that the adhesion and lateral migration of neutrophils themselves did not trigger Rac1 activity in endothelial cells. Together, these findings reveal that JMPs depend on Rac1 activity.

## Targeted exogenous endothelial Rac1 activation promotes leukocyte transmigration

At this point, the contribution of the two neighboring endothelial cells to the JMP is unclear. To study individual, neighboring endothelial cells in more detail, we mixed HUVECs expressing either mNeonGreen-CAAX or mScarlet-I-CAAX. Mosaic-like expression enabled us to distinguish individual endothelial cells that were adjacent to one another (*Figure 6—figure supplement 1A*). Interestingly, LLSM showed the presence of asymmetrical JMPs, meaning that one cell generated the protrusion, while the neighboring cell did not (*Figure 6A* and *Figure 6—figure supplement 1B*). Using scanning electron microscopy, we confirmed that JMPs displayed an asymmetric phenotype at endothelial cell-cell junction regions (*Figure 6B* and *Figure 6—figure supplement 1C*). To quantify this, we analyzed membrane dynamics at JMP regions using mosaic-expressing endothelial cells and found that one endothelial cell showed higher JMP dynamics compared to the adjacent one, as indicated by a ratio >1 (*Figure 6C*). Thus, our data reveal that local JMPs have an asymmetric phenotype at junction regions.

To further assess if local Rac1 activation can trigger the direction of neutrophil transmigration across an asymmetric junction, we overexpressed the Rac-specific RhoGEF Tiam1 in endothelial cells. Biochemical analysis showed that the active Tiam1 mutant, C1199, activated endogenous Rac1 in endothelial cells (*Figure 6—figure supplement 1D* and *Figure 6—source data 1*). Additional immunofluorescence imaging revealed that Tiam1 localized at junctions as determined by a perpendicular line scan across the cell-cell junction indicated by VE-cadherin (*Figure 6—figure supplement 1E–F*). Similar line scan analysis showed that Tiam-C1199 recruited F-actin to these sites (*Figure 6—figure supplement 1G*), in line with previous work by Lampugnani and co-workers (*Lampugnani et al., 2002*). Where endothelial cells that expressed only control plasmids showed low levels of F-actin at junction regions, endothelial cells that showed asymmetric expression of Tiam1 (meaning one cell expressed Tiam1 and the adjacent cell did not) or both endothelial cells expressed Tiam1 showed increased F-actin content at junction regions (*Figure 6—figure supplement 1H–I*). Tiam1-C1199-induced junctional protrusions were approximately 2.5 µm in height, as was determined by 3D imaging (*Figure 6D*) and were reduced in height upon treatment with the Rac1 inhibitor EHT1864

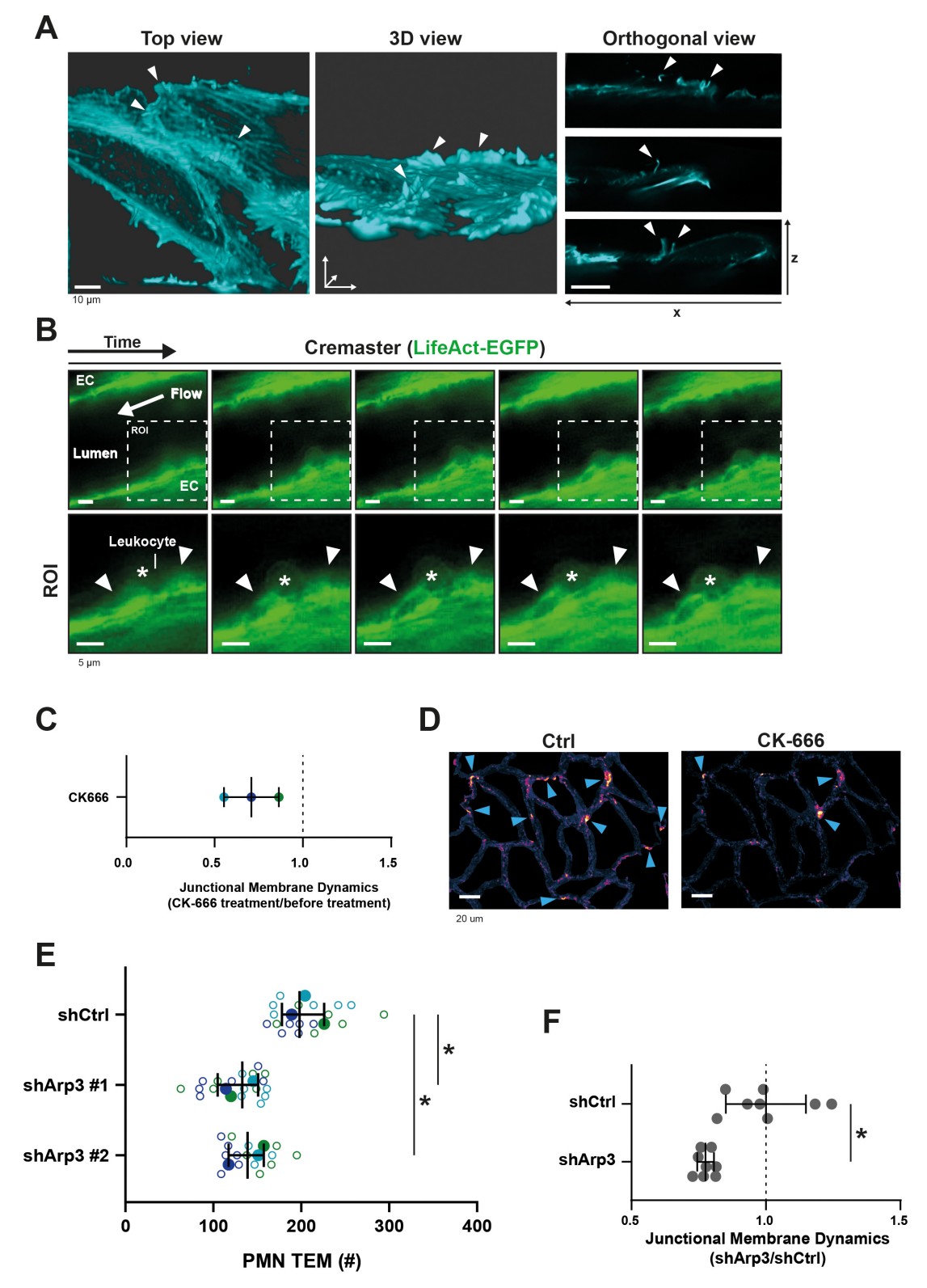

**Figure 4.** JMP regulation by the actin cytoskeleton. (**A**) Stills from two ECs expressing LifeAct-mTurquoise2 showing F-actin-containing JMPs as indicated by white arrowheads. Bar, 10 μm. Images at the right show orthogonal view (XZ direction). (**B**) Confocal intravital microscopy of cremasteric venules in Lifeact-EGFP mice showing the vasculature after four hours with IL-1β and TNFα stimulation. Leukocyte is indicated by asterisk. Arrowheads show existence of JMPs in vivo. Lower panels show region of interest (ROI) zoom. Bar, 5 μm. (**C**) JMP dynamics are measured on HUVECs stimulated

*Figure 4 continued on next page*

*Figure 4 continued*

with TNFα that are treated with the Arp2/3 inhibitor CK-666. Ratio of JMP dynamics after/before CK-666 treatment was calculated. Data points from three independent experiments are shown in three different colors. Median with 95% CI. (D) Example images of membrane dynamics maps upon CK-666 treatment. Control image is from same cells before CK-666 treatment. Blue arrow heads indicate JMPs. Bar, 20 µm. (E) Silencing endothelial Arp3 with two independent shRNAs reduces number of TEM events. Open dots are individual data points from three independent experiments, represented by three different colors. Filled dots are means from three experiments. Median with 95% CI is shown. T-test: Ctrl/sh#1 p=0.0288, Ctrl/sh#2 p=0.0079. (F) Silencing endothelial Arp3 reduces CAAX-positive JMP dynamics. Median with 95% CI is shown. Mann-Whitney U-test: *p=0.0002.

The online version of this article includes the following source data and figure supplement(s) for figure 4:

**Source data 1.** Arp3 knockdown.
**Figure supplement 1.** Arp3-mediated JMPs.

(*Figure 6—figure supplement 2A*). When measuring the direction of migrating neutrophils that crossed the endothelium, we observed that neutrophils preferred to migrate underneath a Tiam1-C1199 cell (*Figure 6E* and *Figure 6—video 1*). Similar results were found when expressing the constitutive active variant of Rac1, Rac1-Q61L, in a mosaic manner with control cells expressing LifeAct (*Figure 6F* and *Figure 6—video 2*). These data show that neutrophils prefer to transmigrate across junctions that display an asymmetric protrusion on one of the two neighboring endothelial cells. As a result, neutrophils migrate from the top of an 'inactive' cell to underneath the 'active' protruding endothelial cell.

Based on these results, we hypothesized that local activation of Rac1, that is, one endothelial cell but not the neighboring one, triggers asymmetric JMPs that function as a local recognition site to drive diapedesis for neutrophils. To test this hypothesis, we used a genetically encoded photoactivatable Rac1 probe (*Wu et al., 2009*). By exciting the Lov2 domain, active Rac1 is released and induced local membrane protrusions (*Figure 6—figure supplement 2B–D*). We found that neutrophils preferred to transmigrate in the direction from the non-transfected cell underneath the PA-Rac1 expressing cell, in line with data presented above (*Figure 6G* and *Figure 6—video 3*). Moreover, local activation of the Rac1 probe increased the number of neutrophils that crossed the endothelium (*Figure 6H* and *Figure 6—figure supplement 2E*).

In summary, our data indicate that neutrophils preferred to exit the endothelium through junctions that display JMPs in an asymmetric manner, meaning that one endothelial cell exposes an apical protrusion and the other one does not (*Figure 6I*). These sites can be recognized as local endothelial TEM hotspots.

## Discussion

Upon acute inflammation, neutrophils start to adhere to the inflamed inner layer of the vessel wall, the endothelium, followed by crawling behavior. When crawling, neutrophils appear to search for an optimal spot to cross the endothelium. Although there is consensus that so-called TEM hotspots exist, and are in particularly recognized in vivo (*Proebstl et al., 2012*; *Hyun et al., 2019*; *Schimmel et al., 2017*), it is unclear how such specific exit-spots are defined and regulated. We found that neutrophils prefer specific, apical membrane structures generated by endothelial cells, serving as exit-sites on inflamed endothelial monolayers.

As leukocytes cross the endothelial barrier within minutes, one requires high resolution and high-speed imaging technology to be able to capture all details in three spatial dimensions in time. To achieve this, we used lattice light sheet microscopy and discovered the existence of endothelial membrane structures that protrude apically at junction regions and serve as local recognition sites for crawling neutrophils. This discovery explains why neutrophils prefer one junction over the other, namely the junction that displays an apical lamella that functionally guides the crawling neutrophil through the junctional cleft underneath the endothelial layer. Hence, such recognition sites may be considered as TEM hotspots. As the membrane protrusion localizes at the junctions, we decided to name this structure the junctional membrane protrusion (JMP). Our data suggest that the JMPs open the junctional cleft and serve as a trap for the crawling neutrophil. JMPs are induced on one endothelial cell whereas the adjacent one shows low lamella activity. Consequently, a local asymmetric protrusion at junction regions appears that intercepts crawling neutrophils (*Figure 6I*). Thus, we

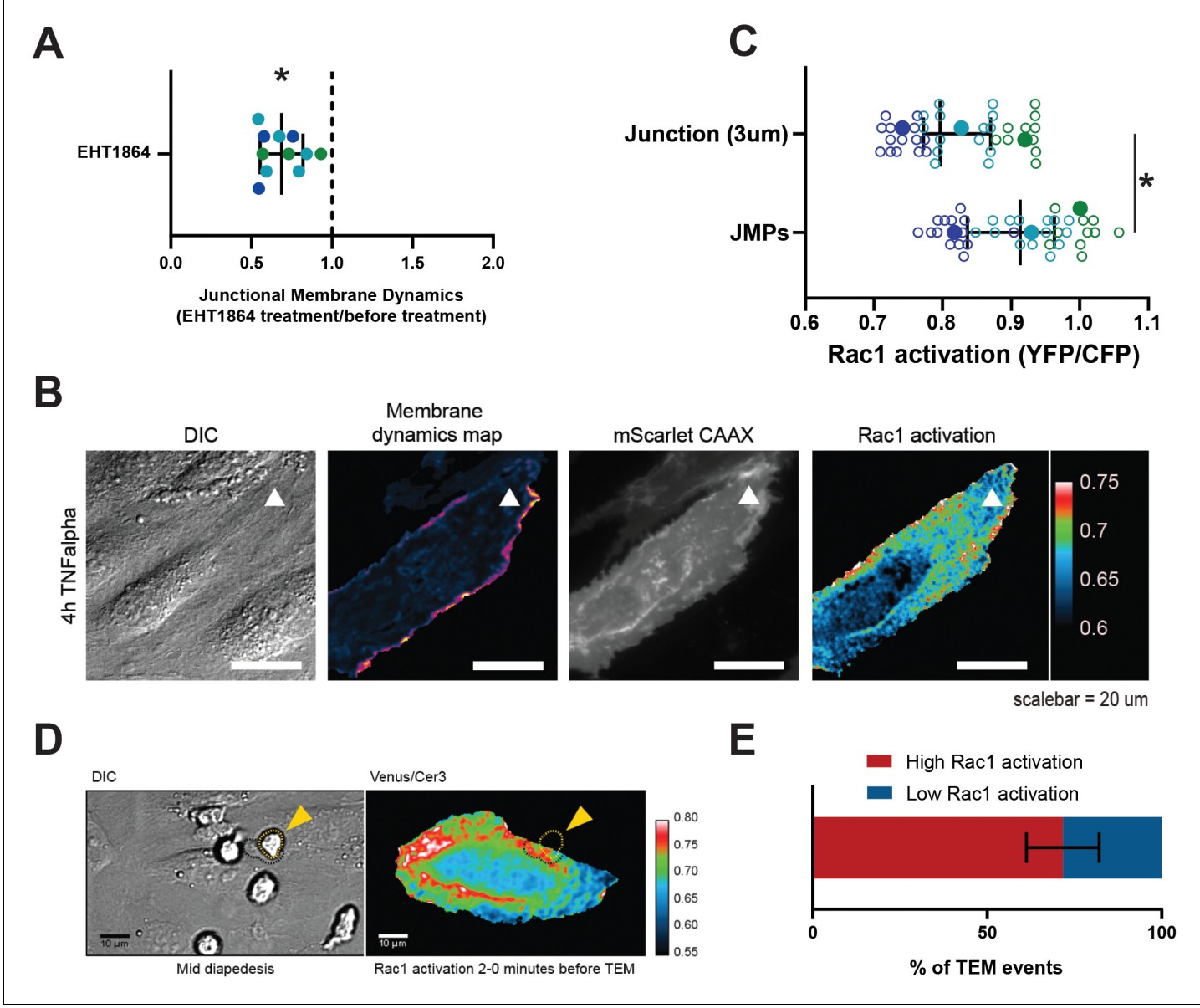

**Figure 5.** JMPs show high Rac1 activity and function as diapedesis hotspot for neutrophils. (**A**) JMP dynamics are measured on HUVECs stimulated with TNFα that are treated with the Rac1 EHT1864 inhibitor. Ratio of JMP dynamics after/before EHT-1864 treatment was calculated. Data points from three independent experiments, represented with three different colors, are shown. Median with 95% CI. One-sample Wilcoxon test: *p<0.0001. (**B**) DIC ECs (i), membrane dynamics map in pseudo colors (ii), mScarlet-I-CAAX membrane label (iii), FRET-based Rac1 biosensor Venus/Cer3 pseudo color ratio-image (iiii) show that JMPs are correlated with high Rac1 activity (white arrowhead). Scale bar, 20 µm. Calibration bar on the right shows high FRET values in warm colors (red) and low FRET values in cold colors (blue) (**C**) Quantification of FRET-based Rac1 biosensor activation at JMPs, selected using mScarlet-I-CAAX, compared to the full junction region of 3 µm wide. Open dots are individual data points from three independent experiments, represented with three different colors. Filled dots are means from three experiments. Median with 95% confidence interval (CI) is shown. T-test: *p=0.0087. (**D**) Still from time-lapse showing PMN TEM under flow. Left image shows DIC of PMN at mid-diapedesis, indicated with yellow arrowhead. Right image shows FRET-based Rac1 biosensor pseudo-color ratio-image. Yellow dotted line indicates part of PMN at the luminal side. Dark dotted line indicates part of PMN at basolateral side. Scale bar, 10 µm. (**E**) Graph shows quantification of PMNs that transmigrate at high Rac1 regions (red bar) versus low Rac1 regions (blue bar).

The online version of this article includes the following video and figure supplement(s) for figure 5:

**Figure supplement 1.** Endogenous Rac1 activation promotes asymmetric JMPs.

**Figure 5—video 1.** Neutrophils prefer spots of high Rac1 activity to transmigrate.

https://elifesciences.org/articles/66074#fig5video1

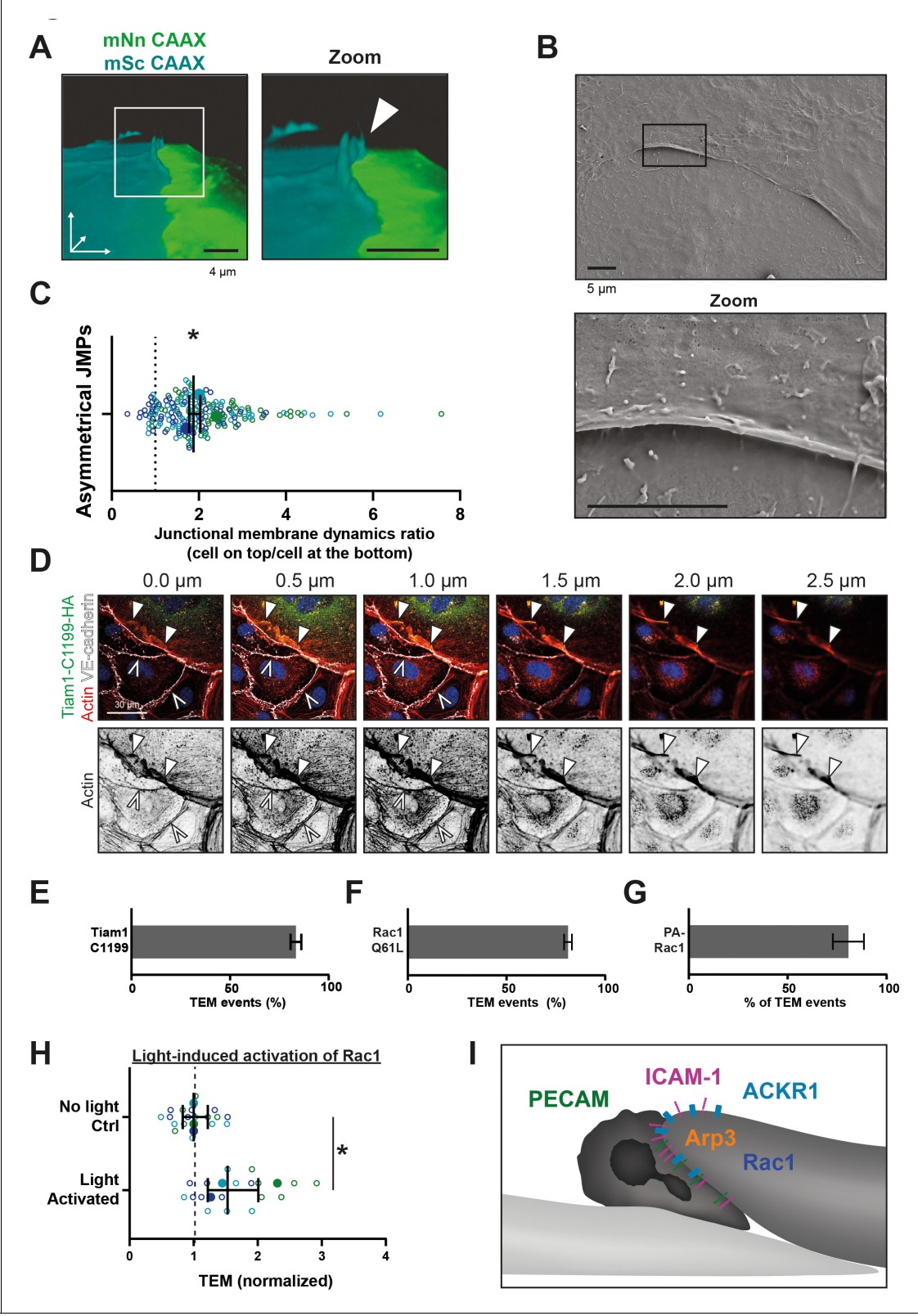

**Figure 6.** Asymmetric JMPs induced by Rac1 and serve as diapedesis sites. (**A**) 3D view still from two endothelial cells showing a turquoise expressing EC and a NeonGreen expressing EC. White box indicates zoom on the right. Arrowhead indicates shows that the turquoise EC displays a membrane protrusion at the junction region, hence, an asymmetric JMP. Bar, 4 μm. (**B**) Scanning electron microscopy image of asymmetric JMPs. Black box indicates zoom region, displayed on the right. Bar, 5 μm. (**C**) Ratio of membrane dynamics in EC with JMP and the other EC at a junction region. Open

*Figure 6 continued on next page*

*Figure 6 continued*

dots are individual data points from three independent experiments, represented by three different colors. Filled dots are means from three experiments. Median with 95% confidence interval (CI) is shown. One-sample T-test: *p=0.344. (D) Immunofluorescent staining for HA (green), F-actin (red), VE-cadherin (white) and DNA (blue) on HUVECs transfected with Tiam1-C1199-HA after overnight TNFα stimulation. Filled white arrowheads indicate the F-actin present at the cell-cell junction between a Tiam1-C1199-HA expressing cell and a control cell, where the open white arrowheads indicate the F-actin present at the junction between two control cells. Panel shows Z-stack from basal (0.0 μm) towards apical (2.5 μm) from left to right, respectively. Arrowheads indicate presence of F-actin rich membrane ruffles in the different focal planes. Scale bar, 30 μm. (E–G) Quantification of diapedesis direction of PMN upon TEM under flow. Majority of the PMNs cross form a wt EC underneath a (E) TIAM1-C1199, (F) Rac1-Q61L, (G) Photoactivatable (PA)-Rac1 EC (dark bar) (H) Quantification of PMN TEM under flow in cells expressing PA-Rac1 either not illuminated (Ctrl) or blue-light illuminated (activated) HUVEC. Data is normalized to control conditions. Upon Rac1 activation, increased number of neutrophils crossed the EC monolayer. Open dots are individual data points from three independent experiments, represented by three different colors. Filled dots are means from three experiments. Median with 95% CI is shown. T-test: p=0.0483. (I) Schematic overview of neutrophil transmigrating from 'inactive' EC (bottom left, light) underneath an 'active' EC (upper right, dark) showing presence of a JMP. JMP displays Rac1 activity, Arp3, and expression of PECAM-1, ICAM-1, and ACKR1.

The online version of this article includes the following video, source data, and figure supplement(s) for figure 6:

**Source data 1.** Tiam-mediated Rac activation.

**Figure supplement 1.** JMPs and exogenous Rac1 activation.

**Figure supplement 2.** JMPs and exogenous Rac1 activation.

**Figure 6—video 1.** Neutrophils prefer spots of Tiam1-induced JMPs to transmigrate.

https://elifesciences.org/articles/66074#fig6video1

**Figure 6—video 2.** Neutrophils prefer to migrate across junctions that asymmetrically display JMP.

https://elifesciences.org/articles/66074#fig6video2

**Figure 6—video 3.** Neutrophils prefer to migrate across junctions that asymmetrically display JMP.

https://elifesciences.org/articles/66074#fig6video3

have identified a unique mechanism that is used by crawling immune cells to efficiently cross the inflamed endothelial barrier in a paracellular manner.

There is ample evidence in the literature indicating that TEM 'hotspots' exist (*Hyun et al., 2019*; *Proebstl et al., 2012*; *Rigby et al., 2015*). However, it has been a longstanding question in the field how such endothelial hotspots are regulated or what would characterize such hotspots. Many options have been proposed to mediate hotspots. For example, local stiffness has been proposed to control this, either induced by the substrate underneath the endothelium or by the endothelium itself by expressing local stiff actin-based structures (*Martinelli et al., 2014*; *Schaefer et al., 2014*). Also local expression of adhesion molecules has been suggested as one of the key elements for such TEM hotspots (*Yang et al., 2005*). And recently, the display of chemokines at junctions has been an attractive hypothesis for the induction of local exit (*Girbl et al., 2018*; *Pruenster et al., 2009*; *Thiriot et al., 2017*). The work by Girbl and colleagues showed that during TEM, endothelial-derived chemokines followed by neutrophil-derived chemokines are presented on the endothelial surface, at junction regions with the help of ACKR1, to guide the neutrophils through. Our data add to this study by showing that JMPs express ACKR1, as well as ICAM-1 and PECAM-1. We propose that these guiding molecules need JMPs for optimal exposure to the crawling leukocytes to initiate diapedesis.

Many different types of membrane structures have been described to emerge in endothelial cells. Breslin and colleagues described local lamellipodia to be involved in controlling the endothelial barrier function (*Breslin et al., 2015*). These structures depend on Rac1 signaling and myosin-mediated local tension and show great similarities to JMPs. JMPs are also regulated by the actin cytoskeleton to the same extent as regular protrusions are induced. We show a prominent role for actin polymerization, Arp2/3-mediated branching and involvement of the small Rho GTPase Rac1. When measuring Rac1 activity locally using a FRET-based biosensor, we find Rac1 to be activated at junction regions of inflamed endothelial cells, and to co-localize with JMPs. When locally activating Rac1, using a photo-activatable Rac1 probe, we can drive leukocyte exit on demand. This is, to the best of our knowledge, the first time that leukocyte extravasation can be triggered on command. Membrane protrusions have also been detected in vivo. Such protrusions were found to be positive for PECAM-1 and surrounded adhering and transmigrating leukocytes (*Petri et al., 2011*; *Phillipson et al., 2008*). These studies argued that the protrusions may function to prevent vascular leakage. Although we did not study if JMPs contribute to the limitation of vascular leakge during leukocyte

emigration, this is an attractive hypothesis. For therapeutic potential, not only can this mechanism now function as a target and pharmacological blockers can be developed, but more importantly, it may lead to novel strategies to initiate leukocyte traffic to parts of the body where we wish to have more immune cells present, for example for immune cell therapies.

For the regulation aspect: it is important to note that JMPs do not seem to be induced by leukocytes themselves, but rather through an intrinsic mechanism within the endothelium triggered by inflammation signals. Indeed, TNFα results in long-term activation of Rac1 (*Van Rijssel et al., 2013*; *Wójciak-Stothard et al., 1998*). Our data does not support a model in which crawling leukocytes trigger the activity or dynamics of JMPs. When measuring JMP dynamics in the presence or absence of crawling leukocytes, we did not detect any difference in JMP activity. Also, the localization of the JMPs did not change when neutrophils were added, and Rac1 activity did not change when leukocytes crawled on top of an endothelial cells. In addition, we noticed that neutrophils did not per se choose the site of highest Rac1 activation in an endothelial cell, presumably because the Rac1 activation itself cannot be sensed from a distance. We hypothesize that a neutrophil continues to crawl over the endothelium until it encounters a local JMP, which is associated with high Rac1 activity, which is then used for diapedesis. We postulate that the regulation of JMPs happens in a stochastic manner, in line with previous reports showing the stochastic behavior of endothelial cell monolayers in early immune responses (*Lipniacki et al., 2006*) and expression of for example von Willebrand factor (*Yuan et al., 2016*).

As endothelial cells are constantly exposed to flow conditions, we expected JMPs to be influenced by flow as well. To our surprise, we did not measure any differences in JMP dynamics or distribution in the absence or presence of long-term flow. This was somewhat surprising, as it is well known that endothelial cells drastically adjust their morphology to long-term flow conditions by polarizing in the direction of the flow with increased Rac1 activity at the upstream side of the endothelial cell (*Kroon et al., 2017*; *Liu et al., 2013*; *Tzima et al., 2002*). However, the stochastic regulation of JMPs under inflammatory conditions is independent of any flow direction. Furthermore, as JMPs localize specifically at intact junction regions, we believe that these structures are therefore per definition different from junction-associated intermittent lamellipodia (JAILs) that have been described previously (*Cao et al., 2017*). JAILs are regulated by VEGFR2 and local release of junctional tension whereas JMPs are present as apical structures on top of VE-cadherin-positive junction regions, display several trafficking molecules and are regulated by inflammatory stimuli.

In summary, we have identified an endothelial membrane structure that supports local exiting of crawling leukocytes. Our data show the presence of such structures in vivo as well. If these structures are also involved and perturbed in diseased conditions is not known but is an attractive hypothesis and would give new opportunities to target such pathologies.

## Materials and methods

**Key resources table**

| Reagent type (species) or resource | Designation | Source or reference | Identifiers | Additional information |
|---|---|---|---|---|
| Primary cells (*Homo-sapiens*) | Human Umbilical Vein Endothelial cells (HUVEC) | Lonza | C2519A: 0000633426 | passage 2-5 |
| Primary cells (*Homo-sapiens*) | Human Lung Microvascular Endothelial cells (HMVEC) | Pelo Biotech | PB-CH-150-4011 | passage 4-8 |
| Primary cells (*Homo-sapiens*) | Human Pancreatic Microvascular Endothelial cells (HMVEC) | Pelo Biotech | PB-CH-147-4011 | passage 4-8 |

*Continued on next page*

*Continued*

| Reagent type (species) or resource | Designation | Source or reference | Identifiers | Additional information |
|---|---|---|---|---|
| Primary cells (*Homo-sapiens*) | Human Kidney Glomerulus Microvascular Endothelial cells (HMVEC) | Pelo Biotech | PB-CH-152-5211 | passage 4-8 |
| Primary cells (*Homo-sapiens*) | Human Embryonic Kidney cells 293T (HEK293T) | ATCC | CRL-3216 | passage 10-30 |
| Transfected construct | CAAX mNeonGreen | van Buul lab | | |
| Transfected construct | CAAX mScarlet-I | van Buul lab | | |
| Transfected construct (*Homo-sapiens*) | ICAM-1 mScarlet-I | van Buul lab | | |
| Transfected construct (*Homo-sapiens*) | ACKR1 mNeonGreen | van Buul lab | | |
| Transfected construct | LifeAct mTurquoise2 | van Buul lab | | |
| Transfected construct (*Homo-sapiens*) | Tiam1-C1199 | *Klems et al., 2020* | | |
| Transfected construct | Rac1 DORA biosensor | *Timmerman et al., 2015* | | |
| Transfected construct | Photoactivatable Rac1 | *Wu et al., 2009* | | |
| Recombinant DNA reagent | pLKO sh Ctrl | Merck | MFCD07785395 | |
| Recombinant DNA reagent | pLKO sh Arp3 | Merck | TRCN0000029381 | |
| Recombinant DNA reagent | pLKO sh Arp3 | Merck | TRCN0000029382 | |
| Antibody | anti human CD144 VE-cadherin AF647 (mouse monoclonal) | BD | Cat #561567 | IF (1:100) |
| Antibody | anti human CD31 PECAM AF647 (mouse monoclonal) | BD | Cat #561654 | IF (1:100) |
| Antibody | anti human actin (mouse monoclonal) | Sigma | Cat #A3853 | WB (1:1000) |
| Antibody | anti human Arp3 (mouse monoclonal) | Merck | cat #A5979 | WB (1:1000) |
| Peptide, recombinant protein | Recombinant Human TNF-alpha | Peprotech | Cat #300-01A | 10 ng/ml |
| Peptide, recombinant protein | Recombinant Human IFN-gamma | R&D | Cat #285-IF | 500 ng/ml |
| Peptide, recombinant protein | IL-1beta | Peprotech | Cat #200-01B | 10 ng/ml |
| Biological compound | LPS | Sigma | Cat #L2880 | 500 ng/ml |

*Continued on next page*

*Continued*

| Reagent type (species) or resource | Designation | Source or reference | Identifiers | Additional information |
|---|---|---|---|---|
| Chemical compound | Lenti-X concentrator | TaKaRa | Cat #631232 | |
| Chemical compound | CK-666 (Arp2/3 inhibitor) | Sigma | Cat #SML0006 | 100 uM |
| Chemical compound | EHT 1864 (Rac inhibitor) | Tocris | Cat #3872 | 50 uM |

## Animals

Lifeact-EGFP transgenic mice have been previously described (*Riedl et al., 2010*). C57BL/6 mice were obtained from Jackson Laboratory. All animal experiments were conducted in accordance with German federal animal protection laws and were approved by the Bavarian Government (Regierung von Oberbayern, Munich, Germany).

## Immunohistochemistry and in vivo microscopy of skeletal muscle

Microsurgical preparation of the cremaster muscle and in vivo microscopy was performed as described previously (*Rehberg et al., 2010*). Briefly, mice were anesthetized using a ketamine/xylazine mixture (100 mg/kg ketamine and 10 mg/kg xylazine), administrated by intraperitoneal injection. The right cremaster muscle was exposed through a ventral incision of the scrotum. The muscle was opened ventrally in a relatively avascular zone, using careful electrocautery to stop any bleeding, and spread over the pedestal of a custom-made microscopy stage. Epididymis and testicle were detached from the cremaster muscle and placed into the abdominal cavity. Throughout the procedure as well as after surgical preparation during in vivo microscopy, the muscle was superfused with warm buffered saline.

After in vivo microscopy, the tissue was fixed in 2% paraformaldehyde and immunostained as whole mount. After washing three times with PBS for 15 min, cells were embedded in Fluoromount (Southern Biotech) and analyzed with a Leica SP8X WLL upright confocal microscope (Leica). Images were analyzed offline using LAS AF software (Leica, Germany), ImageJ (NIH), and IMARIS (Oxford Instruments).

## In vivo microscopy

The setup for in vivo microscopy was centered on an AxioTech-Vario 100 Microscope (Zeiss), equipped with LED excitation light (Zeiss) for fluorescence epi-illumination. Microscopic images were obtained with a water dipping objective (20x, NA 0.5) and acquired with an AxioCam Hsm camera and Axiovision 4.6 software.

## Antibodies

For in vivo analysis, whole mounts of rmTNFα-treated cremaster muscle of C57BL/6 mice were stained with the mouse anti-mouse Coro1A mAb (clone 14.1, SCBT, USA) and the secondary Alexa Fluor donkey 488-conjugated anti-mouse IgG pAb (Thermo Fischer Scientific, USA) and the Alexa Fluor 667-conjugated rat anti-mouse CD31 mAb (clone MEC133, Biolegend, USA).

## Cell culture

Human Umbilical Vein Endothelial Cells (HUVEC) were purchased from Lonza (C2519A:0000633426), cultured in Endothelial Growth Medium two supplemented with singlequots (Promocell) (Cat #C-22011) at 37°C in 5% $CO_2$. Microvascular ECs were purchased from PELO Biotech and cultured in Microvascular EC Growth medium (PELOBiotech) (Cat #PB-MH-100–4099) at 37°C in 5% $CO_2$. All cells were grown in culture flasks, on coverslips and in Ibidi channel slides coated with fibronectin (Sanquin) and used for experiments at passage 4–6. For shear flow experiments in Ibidi slides 0.8 dyne/cm$^2$ was applied.

## Neutrophil transendothelial migration under flow

Polymorphonuclear neutrophils were isolated from whole blood drawn from adult healthy volunteers as described in *Heemskerk et al., 2016* and kept at room temperature for maximally 4 hr prior to the experiment. HUVECs were cultured in a fibronectin-coated Ibidi μ-slide (VI0.4 Ibidi). A perfusion system with HEPES buffer (described before *Heemskerk et al., 2016*) was connected and a shear flow of 0.8 dyne/cm$^2$ was applied. Neutrophils were activated at 37°C for 20–30 min prior to injection into the perfusion system of $1X10^6$ neutrophils. Time-lapse images were recorded for 15–30 min on a widefield microscope with a 40x oil objective at 37°C and 5% $CO_2$. Analysis was performed manually using Fiji.

## Quantification modes of TEM

DIC and fluorescence images were recorded every 5 s. To discriminate from the rolling phase, a neutrophil was crawling when migration speed was below 10 μm/5 s. Implying that its location in two subsequent frames is overlapping. We quantified the number of neutrophils that undergo diapedesis at the first junction they encounter during the crawling phase. We also distinguished between neutrophils that transmigrate at the first region they arrive at a junction and neutrophils that crawled along the junction to their diapedesis site which was defined as being more than 10 μm from the region they initially encountered that junction. Neutrophils that turn around during diapedesis were scored as 'turn' when they were located on top of an endothelial cell adjacent to the endothelial cell they arrived from the frame before diapedesis. To determine whether neutrophils transmigrate at JMPs, we set a threshold at the JMP dynamics map to 2x of the mean. The data to calculate this mean are shown in *Figure 6—figure supplement 2F*. When the diapedesis site had a value above this threshold, it was considered as a diapedesis event at a JMP. These JMP maps were generated from the frames that were collected before the neutrophil arrived at the diapedesis event, to exclude membrane dynamics because of the interaction between the neutrophil and the endothelium. To measure the JMP value at the diapedesis site, a region of interest around the diapedesis site was taken (region 1). To measure the JMP value at junction region a neutrophil first encounters, a region of interest of 10 um was used (region 2). To get the relative MD increase, the difference between these two values was divided by the value of region 2. To measure the relative JMP increase before and after TEM, a region of interest around the diapedesis site was selected. Then the frames before the neutrophils arrived at this site were used for calculating the JMP value (value 1) before TEM. To exclude influences of subendothelial crawling of neutrophils, the 30 frames after a neutrophil left from the diapedesis site were used for calculating the JMP value after TEM (value 2). The difference between these two values was divided by value one to calculate the relative JMP increase upon diapedesis.

## Expression of fluorescent proteins in HUVEC

HUVECs were transduced with lentiviral particles at passage three and seeded in fibronectin-coated Ibidi slides for flow experiments or on coverslips for static experiments. Cells were cultured for 2 days and treated over night with 10 ng/ml TNF-alpha prior to the experiment. For FRET sensor experiments and Rac1 Q61L/LifeAct experiments, cells (either wildtype, or mScarlet-I-CAAX transduced cells) were microporated using the Neon Transfection System (ThermoFisher Scientific) according to manufacturers' protocol and seeded directly in a fibronectin-coated Ibidi slide. The next day, endothelial cells were treated with 10 ng/ml TNF-alpha for 4 hr prior to the experiment.

## Widefield imaging

Widefield images were recorded using a Zeiss Observer Z1 microscope using a 40x NA 1.3 oil immersion objective at 37 degrees with 5% $CO_2$. A HXP 120 V excitation light source at 100% intensity and a Hamamatsu ORCA-R2 digital CCD camera (100–200 msec exposure time for fluorescence, 500 msec exposure time for FRET). Image recordings were done every 2 s (only CAAX recording) or 4–7 s (also DIC and/or FRET recordings).

## JMP maps

HUVECs expressing mNeonGreen-CAAX were imaged on a widefield microscope (Zeiss Observer) with a 40x oil objective for at least 5 min. Fluorescence images were acquired every 2–7 s,

depending on the experiment and the need for DIC recordings. We developed an image-based method for quantifying the dynamics in time series. It is based on the notion that highly dynamic regions will show large fluctuations in intensity. This results in a high standard deviation in the intensity over time. To extract this information, we generated a macro to calculate, for each pixel, the standard deviation in the intensity over time. The standard deviation is normalized by the average intensity and the image essentially depicts the coefficient of variation. Since the intensity fluctuations may arise from sources other than membrane dynamics, we implemented corrections for (i) bleaching and (ii) image drift, resulting in a membrane dynamic map that shows the membrane dynamics of a typical time lapse image of endothelial cells. To focus only on the junctions, the cytoplasm of each cell was selected as being 3 μm from the junctions and set to NaN. To measure junctional membrane dynamics in a specific condition, the mean value of that image was taken. For each experiment, the data was normalized to the condition with HUVEC, cultured on fibronectin-coated glass and treated for 20 hr with TNFα.

## Inhibitors

HUVECs expression mNeonGreen-CAAX to measure membrane dynamics were cultured on coverslips. While imaging 50 μM of EHT1864 or 100 μM CK666 was added. Membrane dynamics were determined 6 min after adding EHT1864 or 20 s after adding CK666. Ratio after versus before inhibitor treatment was calculated from the membrane dynamics values.

## Lattice light sheet microscopy

The lattice light sheet microscope located at the Advanced Imaging Center (AIC) at the Janelia Research Campus of the Howard Hughes Medical Insititute (HHMI) (*Chen et al., 2014*) was used. HUVECs stably expressing mNeonGreen-CAAX or mScarlet-I-CAAX were cultured on fibronectin-coated 5 mm round glass coverslips (Warner Instruments, Catalog # CS-5R) for 2 days. Cells were stimulated with 10 ng/ml TNF-alpha 20 hr prior to imaging. Imaging was performed in HEPES buffer (*Heemskerk et al., 2016*) at 37 degrees with 5% $CO_2$ for maximally 30 min. Neutrophils were isolated as described before/below, stained for 20 min at 37 degrees with Cell Tracker Deep Red (Invitrogen), washed and centrifuged for 3 min, 400G at room temperature and added right above the coverslip between the excitation and detection objectives. 488 nm, 560 nm, and 642 nm diode lasers (MPB Communications) at 30% acousto-optic tunable filter (AOTF) transmittance and 50 mW initial box power and an excitation objective (Special Optics, 0.65 NA, 3.74 mm WD) were used for illumination. Fluorescence was detected via the detection objective (Nikon, CFI Apo LWD 25XW, 1.1 NA) and a sCMOS camera (Hamamatsu Orca Flash 4.0 v2). Exposure time was 20 ms with 50% AOTF transmittance and Z-step size was 0.211 μm. Time interval was about 7.5 s for three channel time-lapse, 5 s for two channel time-lapse and 2.5 s for one channel time-lapse. Point-spread functions were measured using 200 nm tetraspeck beads (Invitrogen cat# T7280) for each wavelength. Data was deskewed and deconvolved as described in Supplemental Methods and analyzed using Imaris software.

## Quantification of JMPs

A semi-automated image analysis pipeline is used to identify JMP regions in the cells imaged using lattice light sheet microscopy. For each frame cell membranes are identified automatically using an Otsu global threshold, and a triangular mesh is created from the volumetric image data using marching cubes iso-surfacing. Local surface variation is measured in the neighborhood of each vertex. JMPs are identified by a global threshold of surface variation. Vertices with high surface variation are considered a JMP region. The threshold was empirically determined and is the same across all datasets. This approach is effective in flat regions of the cells, but is sensitive to the thinner membrane around the nucleus of the cell. Therefore, the nuclear regions were manually marked and ignored for JMP analysis. After identification, JMP regions are separated from the cell mesh and capped to create a closed volume. Area and total region volume are measured in every frame and collected for each dataset. A coverage estimate of the JMP by PECAM or VE-Cadherin stains is also computed by considering the distribution of staining across the JMP regions (details in the supplemental methods). This pipeline was written in MATLAB R2019b (source code can be found on Zenodo at https://doi.org/10.5281/zenodo.5494546; *Arts et al., 2021*).

## Identification of cell membrane surface

The lattice lightsheet used at the Advanced Imaging Center, collects images at a 31.8° angle relative to the coverslip. The images are then 'deskewed' to align each z-slice with one another. In doing so, each z-slice is padded with zeros which can easily throw off any automated segmentation algorithm. To overcome this challenge, a mask is created to determine a valid image region for both the deskewed and deconvolved data. The mask is generated by keeping all non-zero pixels that are identified across the first 50 frames of each dataset. Each z-slice of the mask has hole-filling applied to make sure there are no internal holes.

Within the masked region, a global threshold is estimated for each cell membrane (*Otsu, 1979*). Threshold regions smaller than 0.1% of the masked region are discarded. Regions with convex volume larger than 50% of the mask volume are also discarded. These constraints avoid falsely identifying cell surfaces in highly noisy frames. Following thresholding, a marching cubes isosurface algorithm generates a triangle mesh of the cell surface (*Lorensen and Cline, 1987*).

## Quantification of JMP regions

JMP regions are found in areas of high surface variation. A cylinder is computed for each vertex in the mesh. The cylinder is oriented along the normal of the current vertex with the height and radius of the cylinder such that it contains all vertices in the *n*-neighborhood of the current vertex (for this work $n=20$). Surface variation for each vertex is defined as the volume of the cylinder centered at that vertex. The cell surface mesh is smoothed before computing surface variation, with four iterations of mean smoothing with a 1-neighborhood (vertices directly connected by a mesh edge). This reduces noise to make the calculation of vertex normals and surface variation more robust.

JMP regions are identified by surface variations above a given threshold. In order to provide consistent results across datasets, a global surface variation threshold of 3.5 was chosen empirically. JMP regions are further smoothed using a mesh-based morphological opening with a 1-neighborhood, followed by morphological closing with a 5-neighborhood (*Gonzalez and Woods, 2008*). Finally, holes in the JMP regions are filled.

To identify individual JMPs, each connected region is isolated into sub meshes. Each submesh is closed using an approach similar to *Liepa, 2003*. Once each submesh is capped, the volumes and base areas are measured.

## Coverage of JMP by PECAM/VE-Cadherin

A PECAM or VE-Cadherin stain value is computed for each vertex on the cell surface mesh. The stain value at each vertex is calculated as the mean of the eight voxels surrounding the vertex in the PECAM or VE-Cadherin channel. Vertices are considered 'stained' as long as their stain value is above an Otsu threshold.

To estimate the distribution of stained vertices within a JMP, each vertex in the JMP is also encoded with its minimum surface distance to the edge of the region. A normalized histogram of stained vertices binned by distance from the edge is used to approximate the distribution of PECAM or VE-Cadherin within the JMP region. The normalized histogram is summed to produce an empirical cumulative distribution function. The distance to the 50th, 75th, and 90th percentile of stained vertices is computed. These percentile distances are normalized by the maximum JMP edge distance and averaged over time, per dataset. An illustration of this process for some example staining distributions is shown in *Figure 6—figure supplement 2G*. This gives an estimate of coverage of the JMP by PECAM or VE-Cadherin. If the stain is mostly near the edge, the normalized percentile distances should be near zero. Whereas, if the stain is distributed towards the center of the JMP, then the normalized percentile distances should approach one.

## Scanning electron microscopy

Samples were fixed in 4% paraformaldehyde and 1% glutaraldehyde for 1 hr at room temperature and dehydrated using an ethanol series. To reduce sample surface tension, samples were immersed in hexamethyldisilizane (Sigma-Aldrich) for 30 min and air dried. Before imaging, samples were mounted on aluminum SEM stubs and sputter-coated with a 4 nm platinum-palladium layer using a Leica EM ACE600 sputter coater (Leica Microsystems, Illinois, USA). Images were acquired at 2 kV using a Zeiss Sigma 300 SEM (Zeiss, Germany).

## FRET imaging

Rac1 activation was measured using a DORA FRET-based Rac1 biosensor as described before (*Timmerman et al., 2016*; *Timmerman et al., 2015*). Briefly, we used a Zeiss Observer Z1 microscope equipped with a 40x NA 1.3 oil immersion objective, a HXP 120 V excitation light source, a Chroma 510 DCSP dichroic splitter, and two Hamamatsu ORCA-R2 digital CCD cameras to simultaneously image Cerulean3 and mVenus emission. Data was analyzed using ImageJ software. ROI with no cells presents throughout the movie was selected for background correction of Cerulean3 and mVenus image stacks. Alignment of Cerulean3 and mVenus was done using the 'MultiStackReg' plugin (http://rsb.info.nih.gov/ij/plugins/index.html). To reduce noise a smooth filter was applied to both image stacks. To correct for bleed through, 0.62xCerulean was subtracted from the mVenus signal for each frame to get the corrected mVenus stack, which was divided by Cerulean to calculate the FRET ratio. To define regions of high Rac1 activation, we selected the 1 um junction of the cell and put a threshold to approximately cover 50% of the junction. Then diapedesis sites were scored for either high or low Rac1 activity. To measure Rac1 activation in JMPs, we used the mVenus stacks of the FRET images to select membrane ruffles. These ROIs were projected on the FRET ratio-image and the average FRET value of such a JMP was calculated. Because the FRET signal is always increased at the junctions compared to the cell body, we compared the JMP FRET value to the junction of the cell. For this we selected a border of 3 µm per cell and calculated the mean FRET value.

## Photoactovatable-Rac1 probe

Flow experiment was performed as described above. For the control experiments cells were kept in the dark until neutrophils transmigrated. To switch on photoactivatable Rac-1, channels were illuminated with blue light from a HXP Light source for 10 s prior to neutrophil injection into the perfusion system.

## Reagents

Alexa Fluor 647 mouse anti human CD144 VE-cadherin (55–7 H1) (Cat #561567) and Alexa Fluor 647 mouse anti human CD31 PECAM (WM59) (Cat #561654) were purchased from BD and added live 1:100 to cells 15–60 min prior to imaging. Recombinant Human TNF-alpha (300-01A) was purchased from Peprotech and used at 10 ng/ml for 20 hr (all other experiments) or 4 hr (after microporation) before performing the experiment. Recombinant Human IFN-gamma (R and D) used at 500 ng/ml, IL-1β (Peprotech Cat #200-01B) used at 10 ng/ml and LPS (Sigma) used at 500 ng/ml were added 20 hr before imaging.

Plasmids mNeonGreen-CAAX and mScarlet-I-CAAX were cloned into a lentiviral backbone using HiFi cloning (NEB). shArp3 were purchased from Merck (TRCN0000029381 (CCGGGGCCATGGTA TAGTTGAAGATTCTCGAGAATCTTCAACTATACCATGGCTTTTT), TRCN0000029382 (CCGGCG TCCTCTCTACAAGAATATTCTCGAGAATATTCTTGTAGAGAGGACGTTTTT)). shCtrl (MISSION pLKO.1-puro Non-Mammalian shRNA Control Plasmid DNA Targets no known mammalian genes). mNeonGreen-CAAX was cloned into shArp3 vector (TRCN0000029381) using BamHI and KpnI. Arp3-deficient endorthelial cells showed a normal phenotype with a slightly reduced monolayer resistance and proliferative capacity. Base on VE-cadherin, junctions showed a jagged pattern, comparable to *Mooren et al., 2014*. The DORA Rac1 FRET biosensor (*Timmerman et al., 2016*; *Timmerman et al., 2015*), mCherry Rac1 Q61L (*Klems et al., 2020*) and LifeAct GFP (*Heemskerk et al., 2016*) have been described previously. pTriEx-mCherry-PA-Rac1 (addgene #22027) was a gift from Klaus Hahn. mCherry-PA-Rac1 was digested with NdeI and SmaI and cloned into an empty pLV backbone digested with NdeI and EcoRV, resulting in pLV-mCherry-PA-Rac1. Lentiviral particles were produced in HEK293T cells using 3rd generation packing plasmids. Supernatant was harvested 2 and 3 days after transfection, filtered (0.45 um) and concentrated using Lenti-X Concentrator (TaKaRa). HUVEC transduced with lentiviral particles were used 2–6 days post transduction. Endothelial cells were transduced with the Tiam1-C1199 plasmids, resulting in larger phenotype of the endothelial cells. More details on this phenotype have been described in *Klems et al., 2020*.

## Confocal laser scanning microscopy and image analysis

Immunofluorescent staining was in general performed on HUVECS cultured on 12 mm glass cover-slips coated with 5 µg/ml FN and treated with or without o/n TNFα (10 ng/ml) (Peprotech), washed with PBS+/+(1 mM CaCl2 and 0.5 mM MgCl2), fixated in 4% PFA (Merck), blocked for 30 min with 2% BSA (Affimetrix) and mounted in Mowiol4-88/DABCO solution. Z-stack image acquisition was performed on a confocal laser scanning microscope (Leica SP8) using a 63x NA 1.4 oil immersion objective. 3D reconstruction of Z-stack was made with LasX software (Leica). Junctional actin enrichment was quantified using ImageJ 1.51 p. VE-cadherin labeling was used to visualize cell-cell junctions. 30-pixel (0,18 µm/pixel) width line was drawn on junction of interest and cumulative fluorescent actin signal was measured and normalized by total area. To quantify CAAX, ACKR1 and ICAM-1 signal a Z-stack was acquired. All slices were summed and quantified by plotting the intensity of a line perpendicular to the junction.

## Western blot

Cells expressing shRNA and selected with puromycin were washed once with PBS supplemented with 1 mM $CaCl_2$ and 0.5 mM $MgCl_2$ and lysed with SDS-sample buffer containing 4% b-mercapto-ethanol. Cells expressing shRNA and mNeonGreen-CAAX were sorted using a BD Aria III cell sorter based on mNeonGreen fluorescence. mNeonGreen positive cells were centrifuged (200G, four degrees, 5 min) and lysed with SDS-sample buffer containing 4% b-mercapto-ethanol. Proteins were denatured at 95 degrees for 10 min, separated on a NuPage 4–12% Bis-Tris Gel (Invitrogen) and transferred to a nitrocellulose membrane for 1 hr at 100V. The immunoblots were blocked for 1 hr with 2.5% milk (w/v) in Tris-buffered saline with Tween20 (TBS-T). Primary and secondary antibodies were incubated overnight at four degrees or for 1 hr at room temperature and washed 4 x with TBS-T. Chemiluminescence (Ref# 32106, Thermo Scientific) was detected on light sensitive films (Ref# 47410 19289, Fuji).

## Rac1 activation assay

A confluent monolayer of HUVEC in a 100 mm Petri dish was washed with cold PBS (+ 1 mM CaCl2; 0.5 mM MgCl2) and lysed in 50 mM Tris, pH 7.4, 0.5 mM MgCl2, 500 mM NaCl, 1% (vol/vol) Triton X-100, 0.5% (wt/vol) deoxycholic acid, and 0.1% (wt/vol) SDS supplemented with protease inhibitors. Lysates were cleared at 14,000 $\times$ g for 5 min. GTP-bound Rac1 was isolated with biotinylated Pak1-Crib peptide coupled to streptavidin agarose (*van Buul et al., 2007*). Beads were washed four times in 50 mM Tris, pH 7.4, 0.5 mM MgCl2, 150 mM NaCl, 1% (vol/vol) Triton X-100, and protease inhibitors. Pulldowns and lysates were immunoblotted with monoclonal Rac1 antibodies.

## Acknowledgements

This work was supported by LSBR grant # 1649 (ACIvS), ZonMW NWO Vici grant # 91819632 (JDvB) and PID2019-108902GB-I00 (Spanish Ministry of Science, Innovation and Universities) (EM) and NWO ALW-OPEN grant ALWOP.306 (EKM). The Advanced Imaging Center at Janelia Research Campus is generously sponsored by the Howard Hughes Medical Institute and the Gordon and Betty Moore Foundation.

## Additional information

### Funding

| Funder | Grant reference number | Author |
|---|---|---|
| Landsteiner Foundation for Blood Transfusion Research | 1649 | Abraham CI van Steen |
| NWO | 91819632 | Max LB Grönloh Jaap D van Buul |
| Spanish Ministry of Science and Technology | PID2019-108902GB-I00 | Eloi Montanez |
| NWO | ALWOP.306 | Eike K Mahlandt |

| Deutsche Forschungsge-meinschaft | SFB 914/A02 | Barbara Walzog |
|---|---|---|

The funders had no role in study design, data collection and interpretation, or the decision to submit the work for publication.

## Author contributions

Janine JG Arts, Resources, Data curation, Formal analysis, Validation, Investigation, Visualization, Methodology, Writing - original draft; Eike K Mahlandt, Data curation, Formal analysis, Validation, Investigation, Visualization, Methodology; Max LB Grönloh, Data curation, Formal analysis, Investigation, Methodology; Lilian Schimmel, Data curation, Formal analysis, Methodology; Ivar Noordstra, Abraham CI van Steen, Formal analysis, Investigation, Methodology; Emma Gordon, Resources, Investigation; Simon Tol, Investigation, Methodology; Barbara Walzog, Resources; Jos van Rijssel, Formal analysis, Investigation, Visualization; Martijn A Nolte, Data curation, Formal analysis, Supervision, Writing - review and editing; Marten Postma, Software, Validation; Satya Khuon, Resources, Data curation; John M Heddleston, Resources, Formal analysis, Validation, Methodology; Eric Wait, Resources, Software, Formal analysis, Validation; Teng Leong Chew, Resources, Software, Formal analysis, Validation, Investigation; Mark Winter, Resources, Software, Formal analysis, Validation, Methodology; Eloi Montanez, Data curation, Formal analysis, Visualization; Joachim Goedhart, Formal analysis, Supervision, Funding acquisition, Investigation, Writing - review and editing; Jaap D van Buul, Conceptualization, Resources, Formal analysis, Supervision, Validation, Investigation, Visualization, Writing - original draft, Project administration, Writing - review and editing

## Author ORCIDs

Max LB Grönloh ![iD] https://orcid.org/0000-0003-0109-8225
Lilian Schimmel ![iD] http://orcid.org/0000-0002-0569-0464
Jos van Rijssel ![iD] http://orcid.org/0000-0001-8077-1371
Mark Winter ![iD] http://orcid.org/0000-0003-1180-1957
Eloi Montanez ![iD] http://orcid.org/0000-0003-4059-5056
Joachim Goedhart ![iD] http://orcid.org/0000-0002-0630-3825
Jaap D van Buul ![iD] https://orcid.org/0000-0003-0054-7949

## Ethics

Animal experimentation: All animal experiments were conducted in accordance with German federal animal protection laws and were approved by the Bavarian Government (Regierung von Oberbayern, Munich, Germany).

## Decision letter and Author response

Decision letter https://doi.org/10.7554/eLife.66074.sa1
Author response https://doi.org/10.7554/eLife.66074.sa2

# Additional files

## Supplementary files

• Transparent reporting form

## Data availability

All data generated or analyzed during this study are included in the manuscript and supporting files. Source data files have been provided for Figures 4 and 6.

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
