## [Decision Letter]

**Acceptance summary:**

How neutrophils transmigrate across endothelial cells remains a significant area of research. In this study, Arts and colleagues, propose a novel mechanism that helps to determine exit sites through the endothelial barrier. Specifically, they demonstrate that the formation of junctional membrane protrusions and the underlying activity or presence of Rac1 and Arp3 enhance the likelihood of these junctions to serve as exit sites for neutrophil diapedesis. These important findings should be of interest for the scientific community that focuses on processes of migration and inflammation.

**Decision letter after peer review:**

Thank you for submitting your article "Endothelial junctional membrane protrusions serve as hotspots for neutrophil transmigration" for consideration by *eLife*. Your article has been reviewed by 2 peer reviewers, and the evaluation has been overseen by Carla Rothlin as the Senior and Reviewing Editor. The reviewers have opted to remain anonymous.

Essential revisions:

Overall the manuscript is very strong and we look forward to a revised submission. As you prepare your revised submission, we ask that you please address all the reviewers' comments included below in the "recommendations to authors" and "public reviews" sections.

*Reviewer #2:*

In this manuscript, Arts and colleagues investigated neutrophil transmigration across endothelial cells. They finely characterized the location of neutrophil exit and found that these cells preferentially transmigrate at endothelial cell membrane protrusions at cell junctions (JMPs). They further demonstrate that these endothelial cell membrane protrusions depend on Rac1 activation and they observe preferential neutrophil exit in Rac1 active endothelial cells.

Overall, the manuscript is well written, timely, novel and interesting. This is especially relevant in light of the upcoming possibilities of novel therapeutic strategies to target leukocyte trafficking and induce or block leukocyte extravasation.

*Reviewer #3:*

This study identifies a role for endothelial membrane protrusions at cell junctions as guiding clues for the trans-endothelial migration of neutrophils. It is shown that various inflammatory mediators increase the tendency of endothelial cells to form such protrusions with high dynamics at endothelial junctions. About 70% of transmigrating leukocytes crossed the endothelial barrier at junctions with dynamic membrane protrusions. These protrusions carried PECAM-1, ICAM-1 and the non-signaling chemokine receptor ACKR1, which were reported to be involved in the diapedesis process. A possible equivalent of these protrusions in vivo is demonstrated by PECAM-1 staining in a postcapillary venule of the cytokine-stimulated cremaster muscle. Formation of the membrane protrusions in vitro was dependent on the activity of Arp3 and Rac1. Furthermore, Rac1 activation was observed at such sites based on a FRET-based Rac1 biosensor. PMN diapedesis was preferentially seen at sites of higher Rac1 activity and silencing of Arp3 impaired PMN transmigration. Overexpression of an active form of the Rac1 GEF TIAM1 induced junctional membrane protrusions. Expression of either active Tiam1 or constitutively active Rac1 or photoactivated Rac1 each directed neutrophils to migrate from the top of an "inactive" neighboring cell to underneath the "active" protruding endothelial cell.

This work is important because it demonstrates convincingly that the formation of junctional membrane protrusions and the underlying activity or presence of Rac1 and Arp3 enhance the likelihood of these junctions to serve as exit sites for neutrophil diapedesis. In addition, junctional membrane protrusions seem to facilitate the transmigration of neutrophils in a directed fashion leading to the movement underneath the protruding endothelial cell. Thus, a mechanism is proposed which helps to determine exit sites through the endothelial barrier.

The work would benefit from addressing the following points:

1) It is appreciated that the authors tried to document the existence of junctional membrane protrusions at sites of leukocyte exit in vivo by visualizing neutrophil extravasation in the inflamed cremaster muscle. However, Figure 3D does not allow to determine whether the endothelial membrane protrusions are indeed formed at junctions. In addition, at least for one of the leukocytes that are shown (lower left corner of the depicted vessel segment) it looks like that the endothelial protrusions engulf the leukocytes from two sides. This is highly reminiscent of similar structures that were described in vivo by the Kubes lab several years ago. In these studies, such structures where endothelial cell protrusions engulf the leukocytes had been described as "domes" (Phillipson et al., PLoS one Feb 2008, 3(2):e1649; Petri et al., Blood jan 2011, 117:942-952). These studies suggested that these structures are formed to retain the barrier function of the vessel wall during the diapedesis process. Whether these structures do indeed serve such a protective function is of course difficult to demonstrate. However, the authors of the present study should cite these papers and discuss the point.

In addition, it would need to be shown in the present study, whether the endothelial membrane protrusions found in vivo in cremaster venules are indeed formed at endothelial junctions or whether they may also form at sites of transcellular diapedesis. As it looks in Figure S3B, there are examples depicted where the endothelial membrane protrusions rather extend also from apical sites of the endothelium which are not directly extending from junctions. They very much resemble the "cup" structures, which were described by Olga Barreiro and by Christopher Carman in their in vitro studies. I would also like to add, it has been claimed in reviews in the field that these structures cannot be found in vivo. It is remarkable, how well they are documented here inside postcapillary venules.

2) The study documents that neutrophils migrate preferentially through junctions with high membrane dynamics. This raises the question whether endothelial junctions are usually devoid of membrane protrusions are whether they usually are active in pushing out such protrusions. If endothelial cells form protrusions at junctions for most of the time and at most sites it would be less meaningful that leukocytes prefer such sites, since they are more often faced with such protrusions than not.

3) It is shown that junctional membrane protrusions express PECAM-1, ICAM-1 and ACKR1. It would be useful to know whether these antigens are indeed enriched on junctional membrane protrusions or are expressed at similar levels as at other sites on the cell surface. It is not possible to answer this based on the micrographs that are depicted.

Additional comments

1) The white arrow heads in figures S3E and F seem to be missing.

2) The quantification of fluorescence intensity of ACKR1 and ICAM-1 by line scan analysis depicted in Figure S3D is not well explained. How was the quantification done? Since membrane protrusions go upwards, couldn't that cause stronger fluorescence intensity at such sites, without higher surface density of the antigens?

3) It was not always clear to me, which of the results in the paper were obtained with HUVEC and which with microvascular endothelial cells.

---

## [Author Response]

Reviewer #3:The work would benefit from addressing the following points:1) It is appreciated that the authors tried to document the existence of junctional membrane protrusions at sites of leukocyte exit in vivo by visualizing neutrophil extravasation in the inflamed cremaster muscle. However, Figure 3D does not allow to determine whether the endothelial membrane protrusions are indeed formed at junctions. In addition, at least for one of the leukocytes that are shown (lower left corner of the depicted vessel segment) it looks like that the endothelial protrusions engulf the leukocytes from two sides. This is highly reminiscent of similar structures that were described in vivo by the Kubes lab several years ago. In these studies, such structures where endothelial cell protrusions engulf the leukocytes had been described as "domes" (Phillipson et al., PLoS one Feb 2008, 3(2):e1649; Petri et al., Blood jan 2011, 117:942-952). These studies suggested that these structures are formed to retain the barrier function of the vessel wall during the diapedesis process. Whether these structures do indeed serve such a protective function is of course difficult to demonstrate. However, the authors of the present study should cite these papers and discuss the point.

We thank the reviewer for this and fully agree that these papers should be recognized. We have introduced these papers to our Discussion section at page 25 in the revised manuscript.

In addition, it would need to be shown in the present study, whether the endothelial membrane protrusions found in vivo in cremaster venules are indeed formed at endothelial junctions or whether they may also form at sites of transcellular diapedesis. As it looks in Figure S3B, there are examples depicted where the endothelial membrane protrusions rather extend also from apical sites of the endothelium which are not directly extending from junctions. They very much resemble the "cup" structures, which were described by Olga Barreiro and by Christopher Carman in their in vitro studies. I would also like to add, it has been claimed in reviews in the field that these structures cannot be found in vivo. It is remarkable, how well they are documented here inside postcapillary venules.

The structures in vivo are indeed hard to find. One of the reasons for this is the lack of resolution. We have used real-time imaging on the lattice light sheet microscope to be able to detect and analyze these structures in more detail. Using these tools, we found PECAM-1 to be present on the JMPs but not VE-cadherin. This latter finding indicates that the JMPs are not co-localizing with VE-cadherin but can be detected using PECAM-1. We used PECAM-1 staining in the cremaster and suggest that such structures that we found, after volume rendering of the raw data, may indeed resemble JMPs.

Together with the group of Emma Gordon, Brisbane, Australia, we used the GFP-Lifeact knock in animals and analyzed inflamed cremasters with PECAM-1 stains. Here we could indeed detect a colocalization of PECAM-1 with Lifeact, indicating that the structures are positive for F-actin and at least colocalizing with PECAM-1. This new data set is added to the revised mansucript to supplemental figure 4A.

We agree with the reviewer that we cannot state that the JMPs found in vitro are the same as the ones we detected in vivo. Therefore, we have carefully described the in vivo protrusions and stated that these might resemble JMPs but more future research should reveal the true nature of these in vivo structures. Most likely this includes high resolution imaging. In addition and as described above, the Kubes lab detected PECAM-1-positive structures as well, that may represent JMPs. We have added this information to our Discussion section now.

2) The study documents that neutrophils migrate preferentially through junctions with high membrane dynamics. This raises the question whether endothelial junctions are usually devoid of membrane protrusions are whether they usually are active in pushing out such protrusions. If endothelial cells form protrusions at junctions for most of the time and at most sites it would be less meaningful that leukocytes prefer such sites, since they are more often faced with such protrusions than not.

The protrusions are induced in volume and area size upon inflammatory stimuli, although they are present at control conditions as well, albeit to a limited extend. And as shown in figure 4A, the protrusions are only a few μm in length and most of the time they are smaller. Moreover, supplemental figure 7F shows that only 20% of the junction contains such protrusions. We hypothesize that these sites are preferred exit sites for crawling neutrophils. In this way, potential endothelial damage induced by transmigrating leukocytes is limited to these areas and these sites may help to assist neutrophils to extravasate.

3) It is shown that junctional membrane protrusions express PECAM-1, ICAM-1 and ACKR1. It would be useful to know whether these antigens are indeed enriched on junctional membrane protrusions or are expressed at similar levels as at other sites on the cell surface. It is not possible to answer this based on the micrographs that are depicted.

This is an interesting point raised by the reviewer. We detected these proteins to be present at the protrusions and correlated that with a membrane marker CAAX. We have added a line-scan analysis to these data and found that the increase in protein presence is similar to the increase in membrane marker. This indicated that there was no specific enrichment of one of the proteins detected but rather an increase in membrane density. These data are displayed in figure 3E-F and S3D-G. We noticed that we have not clearly described this issue and have therefore adjusted the text of the result section at page 11 on this topic.

Additional comments1) The white arrow heads in figures S3E and F seem to be missing.

We have added the missing arrowheads.

2) The quantification of fluorescence intensity of ACKR1 and ICAM-1 by line scan analysis depicted in Figure S3D is not well explained. How was the quantification done? Since membrane protrusions go upwards, couldn't that cause stronger fluorescence intensity at such sites, without higher surface density of the antigens?

We apologize that the quantification is not clear. We will add an extra section explaining the quantification in the method section. In short, we made a Z-stack, summed all the slices and then quantified the intensity of the projection. Indeed, it could well be that the increased fluorescence intensity is caused by the accumulation of membrane. This is also the reason why the CAAX signal is increased. We do not state that there is enrichment of ICAM-1 or ACKR1, just that there is colocalization with the increased CAAX signal at the JMPs.

3) It was not always clear to me, which of the results in the paper were obtained with HUVEC and which with microvascular endothelial cells.

We apologize for the confusion. All in vitro experiments were done with HUVECs except Figure S2B, as stated. For all endothelial lines, including microvascular lung endothelial cells, we found the same induction of JMPs upon inflammation.